# SWITCH EMA: A FREE LUNCH FOR BETTER FLATNESS AND SHARPNESS

## ABSTRACT

Exponential Moving Average (EMA) is a widely used weight averaging (WA) regularization to learn flat optima for better generalizations without extra cost in deep neural network (DNN) optimization. Despite achieving better flatness, existing WA methods might fall into worse final performances or require extra test-time computations. This work unveils the full potential of EMA with *a single line of modification*, *i.e.*, switching the EMA parameters to the original model after each epoch, dubbed as Switch EMA (SEMA). From both theoretical and empirical aspects, we demonstrate that SEMA can help DNNs to reach generalization optima that better trade-off between flatness and sharpness. To verify the effectiveness of SEMA, we conduct comparison experiments with discriminative, generative, and regression tasks on vision and language datasets, including image classification, self-supervised learning, object detection and segmentation, image generation, video prediction, attribute regression, and language modeling. Comprehensive results with popular optimizers and networks show that SEMA is a free lunch for DNN training by improving performances and boosting convergence speeds.

## 1 INTRODUCTION

Deep neural networks (DNNs) have revolutionized popular application scenarios like computer vision (CV) (He et al., 2017; Touvron et al., 2021) and natural language processing (NLP) (Devlin et al., 2018) in the past decades. As the size of models and datasets grows simultaneously, it becomes increasingly vital to develop efficient optimization algorithms for better generalization capabilities. A better understanding of the optimization properties and loss surfaces could motivate us to improve the training process and final performances with some simple but generalizable modifications (Wolpert & Macready, 1997; Wallace & Dowe, 1999).

The complexity and high-dimensional parameter space of modern DNNs have posed great challenges in optimization, such as gradient vanishing or exploding, overfitting, and degeneration of large batch size (You et al., 2020). To address these obstacles, two branches of research have been conducted: improving *optimizers* or enhancing optimization by *regularization* techniques. According to their characteristics in Tab. 1, the improved optimizers (Kingma & Ba, 2014; Ginsburg et al., 2018; Zhang et al., 2019; Foret et al., 2021) tend to be more expensive and focus on sharpness(deeper optimal) by refining the gradient, while the popular regularizations (Srivastava et al., 2014; Zhang et al., 2018; Izmailov et al., 2018; Polyak & Juditsky, 1992) are cheaper to use and focus on flatness(wider optimal) by modifying parameters. More precisely, the optimization strategies from both gradient and parameter perspectives show their respective advantages.

Table 1: Comprehensive comparison of optimization and regularization methods from the aspects of pluggable (easy to migrate or not), free gains (performance gain without extra cost or not), speedup (boosting convergence speed or not), and the optimization property (flatness or sharpness).

| Type | Method | Pluggable | Free gains | Speedup | Properties |
|---|---|---|---|---|---|
| Optimizer | SAM | ✓ | ✗ | ✗ | *sharpness* |
| | SASAM | ✗ | ✗ | ✗ | *both* |
| | Adan | ✗ | ✗ | ✓ | *sharpness* |
| | Lookahead | ✓ | ✓ | ✓ | *sharpness* |
| Regularization | SWA | ✓ | ✓ | ✗ | *flatness* |
| | EMA | ✓ | ✓ | ✓ | *flatness* |
| | **SEMA** | ✓ | ✓ | ✓ | ***both*** |

Therefore, a question that deserves to be considered: **is it possible to propose a strategy to optimize both sharpness and flatness simultaneously without incurring additional computational overhead?** Due to simplicity and versatility, the ideal candidate would be weighted averaging (WA)

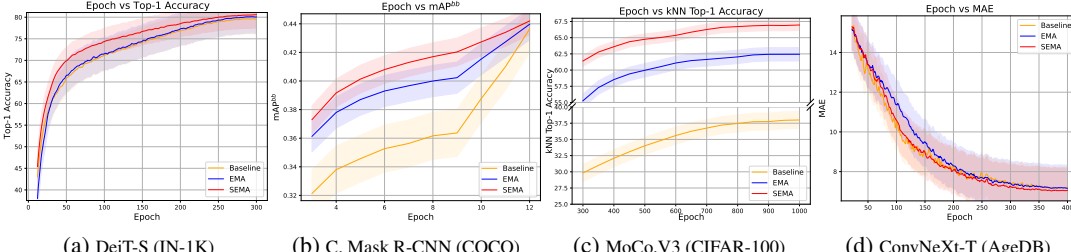

(a) DeiT-S (IN-1K)  (b) C. Mask R-CNN (COCO)  (c) MoCo.V3 (CIFAR-100)  (d) ConvNeXt-T (AgeDB)

Figure 1: Training epoch *vs.* performance plots of the baseline, EMA, and SEMA. (a) Image classification with DeiT-S on ImageNet-1K (IN-1K); (b) Object detection and segmentation with ResNet-50 Cascade (C.) Mask R-CNN (3×) on COCO; (c) Contrastive learning (CL) pre-training with MoCo.V3 and DeiT-S on CIFAR-100; (d) Face age regression with ResNet-50 on AgeDB. SEMA shows faster convergence speeds and better performances than EMA and the baselines.

methods (Izmailov et al., 2018; Polyak & Juditsky, 1992) with vanilla optimizers, widely adopted network regularizers that seek local minima at flattened basins by ensemble model weights. However, previous Weight Averaging (WA) techniques either introduced additional computational overhead, as in the case of TWA (Li et al., 2023c), or operated independently of model optimization, like EMA and SWA, thus maintaining unchanged overall efficiency. The limitations of directly using EMA or SWA during training, which converge quickly but have poor final performance, are underscored by studies like LAWA (Kaddour, 2022) and PSWA (Guo et al., 2022). Techniques like SASAM (Kaddour et al., 2022) indicate that WA can be combined with optimizers to enhance final performance. Consequently, the main objective of this paper is to introduce WA into the optimization process, aiming to expedite convergence while implementing plug-and-play regularization without incurring excessive overhead. In addition to the issue of computational efficiency, we are also inspired by the two-stage optimization strategy of fast and slow: the fast model is used to explore the spiky regions where the empirical risk is minimal (*i.e.,* sharpness), whereas the slow model selects the direction where the risk is more homogeneous (*i.e.,* flatness) for the next update. For example, in regular training utilizing EMA, the fast model corresponds to a model that rapidly updates towards the target in each local iteration, while the slow model precisely aligns with the EMA model. They all have ideal optimization properties but lack the enhancement of interaction during training.

Hence, we introduce *Switch Exponential Moving Average* (SEMA) as a *dynamic regularizer*, which incorporates flatness and sharpness by switching fast and slow models at the end of each training epoch. At each training stage of switching, SEMA fully utilizes the fast convergence of EMA to reach flat local minima, as shown in Figure 1, allowing the optimizer to further explore lower basins through sharp trajectories based on previous EMA parameters for better generalization. In extensive experiments with different tasks and various network architectures, including image classification, self-supervised learning, object detection and segmentation, image generation, regression, video prediction, and language modeling, SEMA improves the performance of baselines consistently as a plug-and-play free lunch. In summary, we make the following contributions:

- We propose the Switch Exponential Moving Average (SEMA) method, and through visualization of the loss landscape and decision boundary experiments, we demonstrate its effectiveness in improving model performance across various scenarios.
- We first apply weight averaging to the training dynamics, allowing SEMA to take both flatness and sharpness into account simultaneously, facilitating faster convergence.
- Comprehensive empirical evidence proves the effectiveness of SEMA. Across numerous popular tasks and datasets, SEMA surpasses state-of-the-art existing WA methods and outperforms alternative optimization methods.

## 2 RELATED WORK

**Optimizers.** With backward propagation (BP) (Rumelhart et al., 1986) and stochastic gradient descending (SGD) (Sinha & Griscik, 1971) with mini-batch training (Bishop, 2006), optimizers play a crucial part in the training process of DNNs. Mainstream optimizers utilize momentum techniques (Sutskever et al., 2013) for gradient statistics and improve DNNs' convergence and performance by adaptive learning rates (*e.g.*, Adam variants (Kingma & Ba, 2014; Liu et al., 2020)) and acceleration schemes (Kobayashi, 2020). SAM (Foret et al., 2021) aims to search a flatter region

where training losses in the estimated neighborhood by solving min-max optimizations, and its variants improve training efficiency (Liu et al., 2022a) from aspects of gradient decomposition (Zhuang et al., 2022), training costs (Du et al., 2021; 2022). To accelerate training, large-batch optimizers like LARS (Ginsburg et al., 2018) for SGD and LAMB (You et al., 2020) for AdamW (Loshchilov & Hutter, 2019) adaptively adjust the learning rate based on the gradient norm to achieve faster training. Adan (Xie et al., 2023) introduces Nesterov descending to AdamW, bringing improvements across popular CV and NLP applications. Another line of research proposes plug-and-play optimizers, *e.g.,* Lookahead (Zhang et al., 2019; Zhou et al., 2021) and Ranger (Wright, 2019), combining with existing inner-loop optimizers (Zhou et al., 2021) and working as the outer-loop optimization.

**Weight Averaging.** In contrast to momentum updates of gradients in optimizers, weight averaging (WA) techniques, *e.g.,* SWA (Izmailov et al., 2018) and EMA (Polyak & Juditsky, 1992), are commonly used in DNN training to improve model performance. As test-time WA strategies, SWA variants (Maddox et al., 2019) and FGE variants (Guo et al., 2023; Garipov et al., 2018) heuristically ensemble different models from multiple iterations (Granziol et al., 2021) to reach flat local minima and improve generalization capacities. TWA (Li et al., 2023c) improves SWA by a trainable ensemble. Model soup (Wortsman et al., 2022) is another WA technique designed for large-scale models, which greedily ensembles different fine-tuned models and achieves significant improvements. When applied during training, EMA update (*i.e.,* momentum techniques) can improve the performance and stabilities. Popular semi-supervised learning (*e.g.,* FixMatch variants (Sohn et al., 2020)) or self-supervised learning (SSL) methods (*e.g.,* MoCo variants (He et al., 2020), and BYOL variants (Grill et al., 2020)) utilize the self-teaching framework, where the parameters of teacher models are the EMA version of student models. In Reinforcement Learning, A3C (Mnih et al., 2016) applies EMA to update policy parameters to stabilize the training process. EMA significantly contributes to the stability and output distribution in generative models like diffusion (Karras et al., 2023). Moreover, LAWA (Kaddour, 2022) and PSWA (Guo et al., 2022) try to apply EMA or SWA directly during the training process and found that using WA during training only accelerates convergence rather than guarantee final performance gains. SASAM (Kaddour et al., 2022) combines the complementary merits of SWA and SAM for better local flatness. Nevertheless, since WA techniques are universal and easy to migrate, they remain crucial for innovation. This perspective introduces WA as a novel approach to the long-unexplored realm of EMA. Our SEMA harnesses the historical data of individual configurations to enhance training efficacy, thereby accelerating convergence rates. Moreover, we leverage the universal applicability of WA methods to bolster EMA's generalization across a spectrum of problem domains, ensuring robust performance across varied scenarios.

**Regularizations.** Network parameter regularizations, *e.g.*, weight decay (Andriushchenko et al., 2023), dropout variants (Srivastava et al., 2014; Huang et al., 2016), and normalization techiniques (Peng et al., 2018; Wu & Johnson, 2021), control model complexity and stabilities to prevent overfitting and are proven effective in improving model generalization. The WA algorithms also fall into this category. For example, EMA can effectively regularize Transformer (Devlin et al., 2018; Touvron et al., 2021) training in both CV and NLP scenarios (Liu et al., 2022b; Wightman et al., 2021). Another part of important regularization techniques aims to improve generalizations by modifying the data distributions, such as label regularizers (Szegedy et al., 2016)) and data augmentations (DeVries & Taylor, 2017). Both data-dependant augmentations like Mixup variants (Zhang et al., 2018; Yun et al., 2019; Liu et al., 2022d) and data-independent methods like RandAugment variants (Cubuk et al., 2019; 2020)) enlarge data capacities and diversities, yielding significant performance gains while introducing ignorable additional computational overhead. Most regularization methods provide "free lunch" solutions that effectively improve performance as a pluggable module with no extra costs. Our proposed SEMA is a new "free-lunch" regularization method that improves generalization abilities as a plug-and-play step for various application scenarios.

## 3 SWITCH EXPONENTIAL MOVING AVERAGE

We present the Switch Exponential Moving Average (SEMA) and analyze its properties. In section 3.1, we consider both the performance and landscape of optimizers (*e.g.,* SGD and AdamW) with or without EMA, which helps understand the loss geometry of DNN training and motivates the SEMA procedure. Then, in section 3.2, we formally introduce the SEMA algorithm. We also derive its practical consequences after applying SEMA to conventional DNN training. Finally, in section 3.3, we provide the theoretical analysis for proving the effectiveness of SEMA.

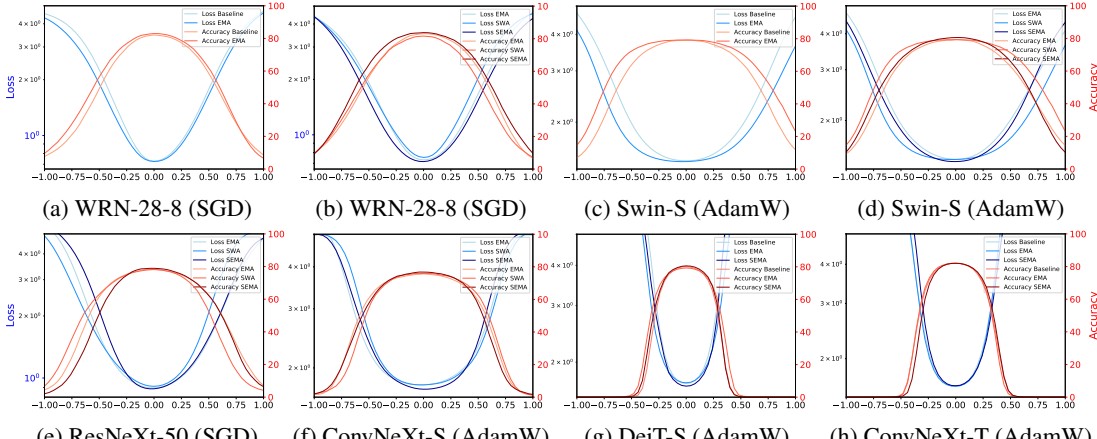

(a) WRN-28-8 (SGD)   (b) WRN-28-8 (SGD)   (c) Swin-S (AdamW)   (d) Swin-S (AdamW)

(e) ResNeXt-50 (SGD)   (f) ConvNeXt-S (AdamW)   (g) DeiT-S (AdamW)   (h) ConvNeXt-T (AdamW)

Figure 2: 1D loss landscape with validation loss (the left axis) and top-1 accuracy (the right axis) of classification on (a)-(f) CIFAR-100 and (g)(h) ImageNet-1K. The loss landscapes of EMA and SWA models are flatter than those of the baseline (using vanilla optimizers), while our proposed SEMA yields deeper and smoother local minima with deepened basins and as flat slopes as EMA. Note that the performance gaps are relatively small on ImageNet-1K due to the massive training data.

### 3.1 Loss Landscape Analysis

SEMA is based on the dynamic weight averaging of switching the slow model generated by EMA to the fast model optimized directly by the optimizer in a specific interval that allows the combination of each unique characteristic to form an intrinsically efficient learning scheme. Therefore, with the popular CNNs and ViTs as backbones, we first analyze the loss landscape and performance to motivate our method.

**EMA.** As a special case of moving average, applies weighting factors that decrease exponentially. Formally, with a momentum coefficient $\alpha \in (0, 1)$ as the decay rate, an EMA recursively calculates the output model weight:

$$\theta_t^{\mathrm{EMA}} = \alpha \cdot \theta_t^{\mathrm{Opt}} + (1 - \alpha) \cdot \theta_{t-1}^{\mathrm{EMA}}, \tag{1}$$

where $\theta^{\mathrm{Opt}}$ represents the model parameters updated by the optimizer, $\theta^{\mathrm{EMA}}$ denotes the exponentially smoothed model parameters, and $t$ is the iteration step in training. A higher $\alpha$ discounts older observations faster.

**Loss Landscape.** The method of visualizing the loss landscape is based on linear interpolation of models (Li et al., 2018) to study the "sharpness" and "flatness" of different minima. The **sharpness** captures the gradient descent's directional sensitivity, and **flatness** assesses the minima's stability for weight averaging. Assuming there is a center point $\theta^*$ as the local minima of the loss landscape and one direction vector $\eta$, the formulation of plotting the loss function $\mathcal{L}$ is:

$$f(\alpha) = \mathcal{L}(\theta^* + \alpha \cdot \eta). \tag{2}$$

For each learned model, the 1-dimensional landscape can be defined by the weight space of the final model. More detailed theoretical explanations are provided in the appendix A.4. In Figure 2, models are trained by optimizers (SGD or AdamW) with or without EMA on CIFAR-100. There are two interesting observations: **(a) the vanilla optimizer without EMA produces a steep peak, whereas (b) with EMA, it has a smoother curve with a lower peak.** These two methods perfectly connect to the two basic properties of loss landscape, flatness, and sharpness, which could be the key to reaching the desired solution of deeper and wider optima for better generalization, while SEMA combines the two advantages without extra computation cost. We further demonstrate the beneficial consequences of using SEMA in the next subsection.

### 3.2 Switch EMA Algorithm

We now present the proposed Switch Exponential Moving Average algorithm, a simple but effective modification for training DNNs. Based on conclusions in section 3.1, since EMA is independent of the learning objective and will stack in the basin without local sharpness, *i.e.,* failing to explore

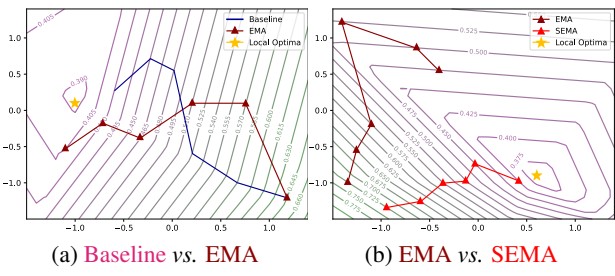

(a) Baseline *vs.* EMA      (b) EMA *vs.* SEMA

Figure 3: Illustration of 2D loss landscape and optimization trajectory on Circles test set. EMA models reach the flat basin while the baseline is stuck at the sharp cliff. Projecting the EMA model to the landscape of SEMA, the SEMA model approaches the local minima efficiently.

**Algorithm 1** Pseudocode for SEMA.

```
# f, f_ema: online and EMA networks
# m: EMA momentum coefficient
f.params = f_ema.params
for i in N epochs: # total epochs
    for x, y in loader: # a minibatch
        loss = LossFunction(f(x), y)
        # optimization and update f.
            params
        loss.backward()
        optimizer.step()
        # momentum update f_ema.params
        f_ema.params = (1-m)*f.params
            + m*f_ema.params
Evaluate(f_ema) # evaluation on
    validation set
# update f.params as f_ema.params
f.params = f_ema.params
```

local minima further. Intuitively, the key issue lies in how to make the slow EMA model $\theta^{\mathrm{EMA}}$ optimizable along with the fast model $\theta^{\mathrm{Opt}}$ during the training process. Therefore, we introduce the simple switching operation between the two models to achieve this goal, *i.e.*, switching $\theta^{\mathrm{Opt}}$ to $\theta^{\mathrm{EMA}}$ regularly according to a predefined switching interval $T$. Formally, SEMA can be defined as:

$$\theta'_t = \theta^{\mathrm{SEMA}}_{t-1} - \eta\nabla\mathcal{L}(\theta^{\mathrm{SEMA}}_{t-1}),$$

$$\theta^{\mathrm{SEMA}}_t = \begin{cases} \theta'_t, & t\%T = 0 \\ \alpha\cdot\theta'_t + (1-\alpha)\cdot\theta^{\mathrm{SEMA}}_{t-1}, & t\%T \neq 0 \end{cases} \tag{3}$$

where $\theta'$ is an intermediate optimizer iterate. Practically, we set $T$ to the multiple of the iteration number for traversing the whole dataset, *e.g.*, switching by each epoch. The training procedure of SEMA is summarized in Algorithm 1, where we only add *a line of code* to the EMA algorithm.

*Three practical consequences of such simple modification on the vanilla optimization process are summarized as follows:*

**Faster Convergence.** SEMA significantly boosts the convergence speed of DNN training. As demonstrated by the 2D loss landscapes in Figure 3, the baseline model frequently gets stuck on the edge of a cliff. In contrast, the EMA model quickly reaches a flat basin. However, when plotted on the SEMA landscape, the model approaches the local minimum via a steeper path, while the EMA model swiftly arrives at a flat, albeit inferior, region. This suggests that SEMA can guide the optimization process towards better solutions, achieving lower losses and reaching the local basin with more efficient strategies and fewer training steps.

**Better Performance.** SEMA enhances the performance of DNNs by skillfully leveraging the strengths of both the baseline and EMA models. SEMA exhibits a deeper and more distinct loss landscape compared to the baseline and existing WA methods, as illustrated in Figure 2b. This starkly contrasts with the EMA, which only shows flatness, and SWA models trained with the straightforward optimizer. This unique characteristic allows SEMA to explore solutions with superior local minima, thereby improving its generalization across various tasks. Intriguingly, SEMA maintains this sharper landscape under different optimizers/backbones 2c. In fact, the loss landscapes 2d of EMA and SWA models appear flatter than that of the baseline.

**Smoother Decision.** SEMA can produce smoother decision boundaries to enhance the robustness of trained models. Figure 4 shows decision boundaries on a toy dataset and illustrates that SEMA models demonstrate greater regularity than EMA models. Conversely, EMA models may have more jagged decision boundaries. The smoother decision boundaries produced by SEMA allow for more reliable and consistent predictions, even in regions with complex data distributions. Figure 2b veri-

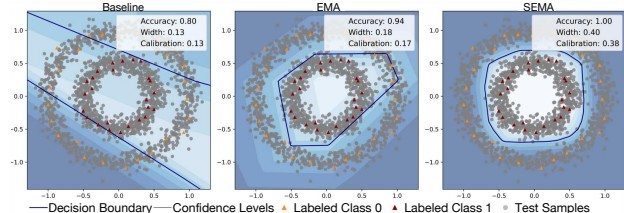

Figure 4: Illustration of the baseline, EMA, and SEMA on Circles Dataset with 50 labeled samples (triangle red/yellow points) and 500 testing samples (gray points) in training a 2-layer MLP. We plot the decision boundary, accuracy, decision boundary width, and prediction calibration.

fies that smoother decision boundaries correspond to better performance with SEMA. This is particularly evident in tasks requiring fine-grained discrimination or complex data distributions because smooth boundaries reduce the chance of overfitting to noisy data or variations. Hence, it ensures more reliable and consistent predictions, further solidifying SEMA's advantage over other methods.

## 3.3 THEORETICAL ANALYSIS

To further substantiate the behavior and advantages of SEMA compared to existing optimization and WA methods in stochastic optimization scenarios and to verify its training stability and convergence characteristics, we take SGD as an example and present a theoretical analysis from two aspects. Let $\eta$ be the learning rate of the optimizer $\alpha$ be the decay rate of SGD, EMA, and SEMA. In the noise quadratic model, the loss of iterates $\Theta_t$ is defined as:

$$\mathcal{L}(\Theta_t) = \frac{1}{2}(\Theta_t - c)^T A (\Theta_t - c), \tag{4}$$

where $c \sim \mathcal{N}(\theta^*, \Sigma)$ and $A$ is the coefficient matrix of the $\mathcal{L}$ with respect to $\Theta$. Without loss of generality, we set $\theta^* = 0$. We denote the models learned by SGD, EMA, and SEMA as $\theta_t^{SGD}$, $\theta_t^{EMA}$, and $\theta_t^{SEMA}$, respectively.

**Proposition 1.** (*Low-frequency Oscillation*): *In the noisy quadratic model, the variance of SGD, EMA, and SEMA iterates, denoted as $V_{SGD}^{(t)} := \mathbb{V}(\Theta_t^{SGD})$, $V_{EMA}^{(t)} := \mathbb{V}(\Theta_t^{EMA})$, and $V_{SEMA}^{(t)} := \mathbb{V}(\Theta_t^{SEMA})$, converge to the following values according to Banach's fixed point theorem, provided $\eta$ satisfies $\frac{2}{\eta} > \lambda_{\max}(A)$, and $V_{SEMA} < V_{EMA} < V_{SGD}$:*

$$V_{SGD} = \frac{\eta A}{2I - \eta A} \Sigma, \quad V_{EMA} = j \cdot V_{SGD}, \quad V_{SEMA} = k \cdot V_{EMA}, \tag{5}$$

where $j < 1$ and $k < 1$ are the coefficients, $j = \frac{\alpha}{2-\alpha} \cdot \frac{2-\alpha-(1-\alpha)\eta A}{\alpha+(1-\alpha)\eta A}$, and $k = \frac{2-\alpha}{\alpha} \cdot \frac{\alpha+(1-\alpha)\eta A}{2-\alpha-(1-\alpha)\eta A} \cdot \frac{\alpha\eta A}{2I-\alpha\eta A} \cdot \frac{2I-\eta A}{\eta A}$. Practically, SEMA's stability can be traced back to its ability to mitigate low-frequency oscillations during optimization. The proposition demonstrates that SEMA achieves a lower variance than EMA and SGD. A lower variance signifies a more stable optimization trajectory, indicating smoother parameter updates and less erratic behavior. As illustrated in Figure 3, SEMA facilitates steady progress towards a local minimum without being impeded by slow and irregular parameter updates. The proof of Proposition 1 is provided in Appendix A.1.

**Proposition 2.** (*Fast Convergence*): *The iterative update of SEMA ensures its gradient descent property as SGD, which EMA doesn't have. It can be formulated as:*

$$(\theta_{t+1}^{SEMA} - \theta_t^{SEMA}) \propto -\nabla\mathcal{L}(\theta_t^{SGD}). \tag{6}$$

Practically, the stability and accelerated convergence of SEMA can be attributed to its ability to integrate the fundamental gradient descent characteristic with rapid convergence. As verified in Figure 3, SEMA's iterative update is proportional to the negative gradient of the loss function. This signifies that SEMA blends the baseline gradient descent characteristic with accelerated convergence, thereby ensuring that the optimization process evolves toward loss reduction and achieves faster convergence. In contrast, EMA does not share the same gradient descent characteristics as SGD. EMA incorporates a smoothing factor that blends current parameter estimates with previous estimates, resulting in a more gradual convergence. Proposition 2 is proofed by Appdenix A.2.

**Proposition 3.** (*Superior Error Bound*): *Building on assumptions and convergence properties of SGD in (Bottou et al., 2016) that considers a fixed learning rate, the error bound of SGD is $\mathcal{E}_{SGD} := \mathbb{E}[\mathcal{L}(\theta^{EMA}) - \mathcal{L}(\theta^*)]$, and error bounds for SGD, EMA, and SEMA can be ranked as, $\mathcal{E}_{SEMA} < \mathcal{E}_{EMA} < \mathcal{E}_{SGD}$:*

$$\mathcal{E}_{SGD} \leq \frac{\eta L M}{2C\mu}, \quad \mathcal{E}_{EMA} \leq \frac{(1-\alpha)\eta L M}{2C\mu}, \quad \mathcal{E}_{SEMA} \leq \frac{\eta L M}{2\sigma_T C\mu}, \tag{7}$$

where $L$ and $C > 0$ are the Lipschitz of $L(\mathcal{L})$ and its constant, $\mu > 1$ and $M > 1$ are the coefficients, and $\sigma_T \geq \frac{\mathbb{E}[\mathcal{L}(\theta_T^{EMA})] - \mathbb{E}[\mathcal{L}(\theta_{2T}^{SEMA})]}{\mathbb{E}[\mathcal{L}(\theta_T^{EMA})] - \mathbb{E}[\mathcal{L}(\theta_{2T}^{EMA})]}$ denotes the improvement of errors by switching once. This proposition further verifies SEMA's strength to exploit a strategic trade-off between SGD and EMA. It switches to SGD at the $T$ iteration to continue optimizing from the sharp landscapes and leverages the smoothness of EMA in $T$ to $2T$ interactions, leading to better error bounds than SGD and EMA. Proposition 3 is proofed by Appdenix A.3.

Table 2: Classification with top-1 accuracy (Acc, %)↑ and performance gains on ImageNet-1K based on various backbones, optimizers, and training epochs (ep). R, CX, and Moga denote ResNet, ConvNeXt, and MogaNet.

| Backbone | R-50 | R-50 | R-50 | R-50 | DeiT-T | DeiT-S | Swin-T | CX-T | Moga-B | DeiT-S | DeiT-B | DeiT-S | DeiT-B | CX-T |
|---|---|---|---|---|---|---|---|---|---|---|---|---|---|---|
| Optimizer | SGD | SAM | LARS | LAMB | AdamW | AdamW | AdamW | AdamW | AdamW | LAMB | LAMB | Adan | Adan | Adan |
| | 100ep | 100ep | 100ep | 300ep | 300ep | 300ep | 300ep | 300ep | 300ep | 100ep | 100ep | 150ep | 150ep | 150ep |
| Basic | 76.8 | 77.2 | 78.1 | 79.8 | 73.0 | 80.0 | 81.2 | 82.1 | 84.5 | 74.1 | 76.1 | 79.3 | 81.0 | 81.3 |
| +EMA | 77.0 | 77.3 | 78.4 | 79.7 | 73.0 | 80.2 | 81.3 | 82.1 | 84.6 | 73.9 | 77.3 | 79.4 | 81.1 | 81.6 |
| **+SEMA** | **77.1** | **77.4** | **78.5** | **79.9** | **73.2** | **80.6** | **81.6** | **82.2** | **84.8** | **74.4** | **77.4** | **79.5** | **81.3** | **81.7** |
| Gains | 0.3 | 0.2 | 0.1 | 0.1 | 0.2 | 0.6 | 0.4 | 0.1 | 0.3 | 0.3 | 1.3 | 0.2 | 0.3 | 0.4 |

Table 3: Classification with top-1 accuracy (%)↑ and performance gains on CIFAR-100 based on various CNN and Transformer backbones.

| Backbone | Basic | +EMA | +SWA | +Lookahead | **+SEMA** | Gains |
|---|---|---|---|---|---|---|
| VGG-13 (BN) | 75.19±0.68 | 75.47±0.15 | 75.30±0.10 | 75.26±0.46 | **75.80**±0.12 | 0.61 |
| R-18 | 76.91±0.43 | 77.16±0.08 | 77.13±0.09 | 77.07±0.75 | **77.61**±0.08 | 0.70 |
| RX-50 | 79.06±0.34 | 79.21±0.07 | 79.25±0.09 | 79.28±0.49 | **79.80**±0.06 | 0.74 |
| R-101 | 76.90±0.31 | 77.48±0.05 | 77.41±0.06 | 77.27±0.15 | **77.62**±0.06 | 0.72 |
| WRN-28-10 | 81.94±0.62 | 82.27±0.12 | 81.16±0.09 | 81.20±1.03 | **82.35**±0.10 | 0.41 |
| DenseNet-121 | 80.49±0.47 | 80.70±0.07 | 80.83±0.05 | 80.74±0.45 | **81.05**±0.08 | 0.56 |
| DeiT-S | 63.34±0.59 | 64.46±0.10 | 64.17±0.09 | 64.25±0.64 | **64.58**±0.09 | 1.24 |
| MLPMixer-T | 78.22±0.46 | 78.49±0.07 | 78.54±0.05 | 78.33±0.37 | **78.84**±0.07 | 0.62 |
| Swin-T | 79.07±0.32 | 79.17±0.08 | 79.30±0.07 | 79.28±0.82 | **79.74**±0.07 | 0.67 |
| Swin-S | 78.25±0.42 | 79.08±0.09 | 78.93±0.06 | 78.76±0.51 | **79.30**±0.09 | 1.05 |
| ConvNeXt-T | 78.37±0.23 | 79.24±0.06 | 78.96±0.08 | 78.82±0.28 | **79.42**±0.06 | 1.05 |
| ConvNeXt-S | 60.18±0.39 | 61.45±0.07 | 61.04±0.07 | 60.29±0.32 | **61.76**±0.09 | 1.58 |
| MogaNet-S | 83.69±0.50 | 83.92±0.09 | 83.78±0.07 | 83.67±1.02 | **84.02**±0.08 | 0.33 |

Table 4: Pre-training with top-1 accuracy (%)↑ of linear probing (Lin.) or fine-tuning (FT) and performance gains on CIFAR-100 and STL-10 based on various SSL algorithms.

| Self-sup | Dataset | Backbone | Basic | +EMA | +SWA | **+SEMA** | Gains |
|---|---|---|---|---|---|---|---|
| SimCLR | CIFAR-100 | R-18 | 67.18±0.85 | 58.46±0.09 | 57.82±0.15 | **67.28**±0.08 | 0.10 |
| SimCLR | STL-10 | R-50 | 91.77±0.36 | 82.82±0.08 | 91.36±0.09 | **92.93**±0.10 | 1.16 |
| MoCo.V2 | CIFAR-100 | R-18 | 62.34±0.83 | 66.53±0.49 | 62.85±0.24 | **66.56**±0.21 | 0.03 |
| MoCo.V2 | STL-10 | R-50 | 91.33±0.27 | 91.36±0.18 | 91.40±0.13 | **91.48**±0.09 | 0.12 |
| BYOL | CIFAR-100 | R-18 | 55.09±0.79 | 69.60±0.20 | 56.36±0.15 | **69.86**±0.13 | 0.26 |
| BYOL | STL-10 | R-50 | 75.76±0.34 | 93.24±0.09 | 76.29±0.10 | **93.48**±0.08 | 0.24 |
| BarlowTwins | CIFAR-100 | R-18 | 65.49±1.07 | 60.13±0.12 | 60.53±0.16 | **65.55**±0.11 | 0.06 |
| BarlowTwins | STL-10 | R-50 | 88.67±0.26 | 80.11±0.09 | 88.35±0.14 | **88.80**±0.09 | 0.13 |
| MoCo.V3 | CIFAR-100 | DeiT-S | 38.09±1.26 | 46.79±0.12 | 39.61±0.34 | **52.27**±0.13 | 5.48 |
| MoCo.V3 | STL-10 | DeiT-S | 61.88±0.30 | 79.25±0.07 | 62.49±0.15 | **80.44**±0.06 | 1.19 |
| SimMIM | CIFAR-100 | DeiT-S | 81.96±0.10 | 82.05±0.07 | 81.77±0.07 | **82.15**±0.08 | 0.19 |
| SimMIM | STL-10 | DeiT-S | 91.88±0.10 | 69.14±0.05 | 78.23±0.06 | **92.06**±0.04 | 0.18 |
| A²MIM | CIFAR-100 | DeiT-S | 82.28±0.15 | 82.14±0.09 | 82.05±0.10 | **82.46**±0.03 | 0.18 |
| A²MIM | STL-10 | DeiT-S | 92.27±0.09 | 70.88±0.08 | 80.64±0.09 | **93.33**±0.07 | 1.06 |

## 4 EXPERIMENTS

### 4.1 EXPERIMENTAL SETUP

We conduct extensive experiments across a wide range of popular application scenarios to verify the effectiveness of SEMA. Taking vanilla optimizers as the baseline (basic), the compared regularization methods plugged upon the baseline include EMA (Polyak & Juditsky, 1992), SWA (Izmailov et al., 2018), and Lookahead optimizers (Zhang et al., 2019). We use the momentum coefficients of 0.9999 and 0.999 for EMA and SEMA, 1.25 budge for SWA, and one epoch switch interval for SEMA. As for the vanilla optimizers, we consider SGD variants (momentum SGD (Sinha & Griscik, 1971) and LARS (Ginsburg et al., 2018)) and Adam variants (Adam (Kingma & Ba, 2014), AdamW (Loshchilov & Hutter, 2019), LAMB (You et al., 2020), SAM (Foret et al., 2021), Adan (Xie et al., 2023). View Appendix B for details of implementations and hyperparameters. All experiments are implemented with PyTorch and run on NVIDIA A100 or V100 GPUs, and we use the **bold** and grey backgrounds as the default baselines. The reported results are averaged over three trials. We intend to verify three empirical merits of SEMA: (i) **Convenient plug-and-play usability**, as the basic optimization methods we compared, SEMA enables convenient plug-and-play as a regularizer; (ii) **Higher generalization performance gain**, SEMA can take into account both flatness and sharpness, which makes it more able to converge the local optimal position than other optimization methods, thus bringing higher performance gains to the model. Relative to baselines, EMA, and other techniques, SEMA achieves higher performance gains and significantly enhances the EMA generalization; (iii) **Faster convergence**, SEMA inherits the fast convergence properties of EMA while benefiting from gradient descent, allowing it to help models converge faster.

### 4.2 EXPERIMENTS FOR COMPUTER VISION TASKS

We first apply WA regularizations to comprehensive vision scenarios that cover discriminative, generation, predictive, and regression tasks to demonstrate the versatility of SEMA on CIFAR-10/100 (Krizhevsky et al., 2009), ImageNet-1K (IN-1K) (Deng et al., 2009), STL-10 (Coates et al., 2011), COCO (Lin et al., 2014), CelebA (Liu et al., 2015), IMDB-WIKI (Rothe et al., 2018), AgeDB (Moschoglou et al., 2017), RCFMNIST (Yao et al., 2022), and Moving MNIST (MM-NIST) (Srivastava et al., 2015) datasets.

**Image classification.** Evaluations are carried out from two perspectives. Firstly, we verify popular network architectures on the standard CIFAR-100 benchmark with 200-epoch training: (a) classical Convolution Neural Networks (CNNs) include ResNet-18/101 (R) (He et al., 2016), ResNeXt-

50-32x4d (RX) (Xie et al., 2017), Wide-ResNet-28-10 (WRN) (Zagoruyko & Komodakis, 2016), and DenseNet-121 (Huang et al., 2017); (b) Transformer (Metaformer) architectures include DeiT-S (Touvron et al., 2021), Swin-T/S (Liu et al., 2021), and MLPMixer-T (Tolstikhin et al., 2021); (c) Modern CNNs include ConvNeXt-T/S (CX) (Liu et al., 2022b) and MogaNet-S (Moga) (Li et al., 2024c). Note that classical CNNs are trained by SGD optimizer with $32^2$ resolutions, while other networks are optimized by AdamW with $224^2$ input size. Table 3 notably shows that SEMA consistently achieves the best top-1 Acc compared to WA methods and Lookahead across 12 backbones, where SEMA also yields fast convergence speeds in Figure 1. Then, we further conduct large-scale experiments on IN-1K to verify various optimizers (*e.g.,* SGD, SAM, LARS, LAMB, AdamW, and Adan) using standardized training procedures and the networks mentioned above. In Table 2, SEMA enhances a wide range of optimizers and backbones, *e.g.,* +0.6/1.3/0.4% Acc upon DeiT-S/DeiT-B/CX-T with AdamW/LAMB/Adan, while conducting Acc gains in situations where EMA is not applicable (*e.g.,* R-50 and DeiT-S with LAMB). View Appendix B.1 for details.

**Self-supervised Learning.** Since EMA plays a vital role in some SSL methods, we also evaluate WA methods with two categories of popular SSL methods on CIFAR-100, STL-10, and IN-1K, *i.e.,* contrastive learning (CL) methods include SimCLR (Chen et al., 2020a), MoCo.V2 (Chen et al., 2020b), BYOL (Grill et al., 2020),

Table 5: Pre-training (PT) with top-1 accuracy (%)↑ of linear probing or FT and performance gains on IN-1K based on SSL methods with various PT epochs.

| Dataset | Backbone | PT | Basic | +EMA | +SWA | +SEMA | Gains |
|---|---|---|---|---|---|---|---|
| BYOL | R-50 | 200ep | 65.49 | 69.78 | 66.37 | **69.96** | 0.18 |
| MoCo.V3 | DeiT-S | 300ep | 67.73 | 71.77 | 68.54 | **72.01** | 0.24 |
| SimMIM | DeiT-B | 800ep | 83.85 | 83.94 | 83.79 | **84.16** | 0.31 |
| MAE | DeiT-B | 800ep | 83.33 | 83.37 | 83.35 | **83.48** | 0.15 |

Barlow Twins (BT) (Zbontar et al., 2021), and MoCo.V3 (Chen et al., 2021), which are tested by linear probing (Lin.), and masked image modeling (MIM) include MAE (He et al., 2022), SimMIM (Xie et al., 2022), and A$^2$MIM using fine-tuning (FT) protocol. Notice that most CL methods utilize ResNet variants (optimized by SGD or LARS) as the encoders, while MoCo.V3 and MIM algorithms use ViT backbones (optimized by AdamW). Firstly, we perform 1000-epoch training on small-scale datasets with $224^2$ resolutions for fair comparison in Table 4, where SEMA performs best upon CL and MIM methods. When EMA is used in self-teaching frameworks (MoCo.V2/V3 and BYOL), SEMA improves EMA by 0.12~5.48% Acc on STL-10 where SWA fails to. When EMA and SWA showed little gains or negative effects upon MIM methods, SEMA still improves them by 0.18~1.06%. Then, we compare WA methods on IN-1K with larger encoders (ResNet-50 and ViT-S/B) using the standard pre-training settings. As shown in Table 5, SEMA consistently yields the most performance gains upon CL and MIM methods. View Appendix B.2 for details.

Table 6: Object detection and segmentation with mAP$^{bb}$ (%)↑, mAP$^{mk}$ (%)↑, and performance gains on COCO based on Mask R-CNN and its Cascade (Cas.) version.

| Method | Basic | | +EMA | | +SEMA | | Gains | |
|---|---|---|---|---|---|---|---|---|
| | AP$^{bb}$ | AP$^{mk}$ | AP$^{bb}$ | AP$^{mk}$ | AP$^{bb}$ | AP$^{mk}$ | AP$^{bb}$ | AP$^{mk}$ |
| Mask R-CNN (2×) | 39.1 | 35.3 | 39.3 | 35.5 | **39.7** | **35.8** | 0.6 | 0.5 |
| Cas. Mask R-CNN (3×) | 44.0 | 38.3 | 44.2 | 38.5 | **44.4** | **38.6** | 0.4 | 0.3 |
| Cas. Mask R-CNN (9×) | 44.0 | 38.5 | 44.5 | 38.8 | **45.1** | **39.2** | 1.1 | 0.7 |

Table 7: Object detection with mAP$^{bb}$ (%)↑ and performance gains on COCO based on various detection methods and different backbones with fine-tuning or training from scratch setups.

| Method | Backbone | Basic | +EMA | +SWA | **+SEMA** | Gains |
|---|---|---|---|---|---|---|
| RetinaNet (2×) | R-50 | 37.3 | 37.6 | 37.6 | **37.7** | 0.4 |
| RetinaNet (1×) | Swin-T | 41.6 | 41.8 | 41.9 | **42.1** | 0.5 |
| YoloX (300ep) | YoloX-S | 37.7 | 40.2 | 39.6 | **40.5** | 0.3 |

**Object Detection and Instance Segmentation.** As WA techniques (Zhang et al., 2020) were verified to be useful in detection (Det) and segmentation (Seg) tasks, we benchmark them on COCO with two types of training settings. Firstly, using the standard fine-tuning protocol in MMDetection (Chen et al., 2019), RetinaNet (Lin et al., 2017), Mask R-CNN (He et al., 2017), and Cascade Mask R-CNN (Cas.) (Cai & Vasconcelos, 2019) are fine-tuned by SGD or AdamW with IN-1K pre-trained R-50 or Swin-T encoders, as shown in Table 6 and Table 7. SEMA achieved substantial gains of AP$^{bb}$ and AP$^{mk}$ over the baseline with all methods and exceeded gains of EMA models. Then, we train YoloX-S detector (Ge et al., 2021) from scratch by SGD optimizer for 300 epochs in Table 7. It takes EMA as part of its training strategy, where EMA significantly improves the baseline by 2.5% AP$^{bb}$, while SEMA further improves EMA by 0.3% AP$^{bb}$. View Appendix B.3 for details.

**Image Generation.** Then, we investigate WA methods for image generation (Gen) tasks based on DDPM (Ho et al., 2020) on CIFAR-10 and CelebA-Align because EMA significantly enhances image generation, es-

Table 8: Image generation with FID (%)↓ and performance gains on CIFAR-10 and CelebA-Align.

| Dataset | Basic | +EMA | +SWA | +Lookahead | **+SEMA** | Gains |
|---|---|---|---|---|---|---|
| CIFAR-10 | 7.17$_{\pm0.18}$ | 5.43$_{\pm0.03}$ | 6.35$_{\pm0.08}$ | 6.84$_{\pm0.12}$ | **5.30**$_{\pm0.06}$ | 0.13 |
| CelebA-Align | 7.90$_{\pm0.21}$ | 7.49$_{\pm0.07}$ | 7.53$_{\pm0.06}$ | 7.67$_{\pm0.23}$ | **7.11**$_{\pm0.07}$ | 0.38 |

pecially with diffusion models. In Table 8, the FID of DDPM drops dramatically without using

EMA, making it necessary to adopt WA techniques. SEMA can yield considerable FID gains and outperform other optimization methods on all datasets. View Appendix B.4 for details.

**Video Prediction.** Employing OpenSTL benchmark (Tan et al., 2023), we verify the video prediction (VP) task on MM-NIST with various VP methods. In Table 9, SEMA can improve MSE and PSNR metrics for recurrent-based (ConvLSTM (Shi et al., 2015) and PredRNN (Wang et al., 2017))

Table 9: Video prediction with MSE↓, PSNR↑, and performance gains on MMNIST upon VP baselines.

| Method | Basic | | +EMA | | +SEMA | | Gains (%) | |
|---|---|---|---|---|---|---|---|---|
| | MSE | PSNR | MSE | PSNR | MSE | PSNR | MSE | PSNR |
| SimVP | 32.15 | 21.84 | 32.14 | 21.84 | **32.06** | **21.85** | 0.28 | 0.05 |
| SimVP.V2 | 26.70 | 22.78 | 27.12 | 22.75 | **26.68** | **22.81** | 0.07 | 0.13 |
| ConvLSTM | 23.97 | 23.28 | 24.06 | 23.27 | **23.92** | **23.31** | 0.21 | 0.13 |
| PredRNN | 29.80 | 22.10 | 29.76 | 22.15 | **29.73** | **22.16** | 0.23 | 0.27 |

and recurrent-free models (SimVP and SimVP.V2 (Gao et al., 2022)) compared to the baseline while other WA methods usually degrade performances. View Appendix B.5 for details.

**Visual Attribute Regression.** We further evaluate face age regression tasks on AgeDB (Moschoglou et al., 2017) and IMDB-WIKI (Rothe et al., 2018) and pose regression on RCFMNIST (Yao et al., 2022) using $\ell_1$ loss. As shown in Table 10, SEMA achieves significant gains in terms of MAE and RMSE metrics compared to the baseline methods on various datasets, particu-

Table 10: Regression tasks with MAE↓, RMSE↓, and performance gains on RCF-MNIST, AgeDB, and IMDB-WIKI based on various backbone encoders.

| Dataset | Back. | Basic | | +EMA | | +SWA | | +SEMA | | Gains (%) | |
|---|---|---|---|---|---|---|---|---|---|---|---|
| | | MAE | RMSE | MAE | RMSE | MAE | RMSE | MAE | RMSE | MAE | RMSE |
| RCF-MNIST | R-18 | 5.61 | 27.30 | 5.50 | 27.70 | 5.53 | 27.73 | **5.41** | **26.79** | 3.57 | 1.79 |
| RCF-MNIST | R-50 | 6.20 | 28.78 | 5.81 | 27.19 | 6.04 | 27.87 | **5.74** | **27.71** | 7.42 | 3.72 |
| AgeDB | R-50 | 7.25 | 9.50 | 7.33 | 9.62 | 7.37 | 9.59 | **7.22** | **9.49** | 0.41 | 0.11 |
| AgeDB | CX-T | 7.14 | 9.19 | 7.31 | 9.39 | 7.28 | 9.34 | **7.13** | **9.13** | 0.14 | 0.65 |
| IMDB-WIKI | R-50 | 7.46 | 11.31 | 7.50 | 11.24 | 7.62 | 11.45 | **7.49** | **11.11** | 0.40 | 1.77 |
| IMDB-WIKI | CX-T | 7.72 | 11.67 | 7.21 | 10.99 | 7.87 | 11.71 | **7.19** | **10.94** | 6.86 | 6.26 |

larly with a 7.42% and 3.72% improvement on the R-50 model trained on the RCF-MNIST dataset and a 6.86% and 6.26% improvement on the CX-T model trained on IMDB-WIKI. Moreover, the experimental results consistently outperform the models trained using EMA and SWA training strategies. View details in Appendix B.6.

Table 11: Languaging processing on Penn Treebank with perplexity↓ based on 2-layer LSTM.

| Optimizer | Basic | +EMA | +SWA | +SEMA | Gains |
|---|---|---|---|---|---|
| SGD | 67.5±0.05 | 67.3±0.05 | 67.4±0.04 | **67.1±0.06** | 0.4 |
| Adam | 67.3±0.04 | 67.2±0.07 | 67.1±0.03 | **67.0±0.05** | 0.3 |
| AdaBelief | 66.2±0.05 | 66.1±0.10 | 66.0±0.09 | **65.9±0.08** | 0.3 |

Table 12: Text classification and languaging modeling with Acc (%)↑ and Perplexity (P)↓ on Yelp Review and WikiText-103 based on BERT-Base.

| Dataset | Metric | Basic | +EMA | +SWA | +SEMA | Gains |
|---|---|---|---|---|---|---|
| Yelp Review | Acc↑ | 68.26±0.45 | 68.35±0.11 | 68.38±0.09 | **68.46±0.10** | 0.20 |
| WikiText-103 | P↓ | 29.92±0.21 | 29.57±0.08 | 29.60±0.07 | **29.46±0.07** | 0.46 |

## 4.3 Experiments for Language Processing Tasks

Then, we also conduct experiments with classical NLP tasks on Penn Treebank, Yelp Review, and WikiText-103 datasets to verify whether the merits summarized above still hold. Following AdaBelief (Zhuang et al., 2020), we first evaluate language processing with 2-layer LSTM (Ma et al., 2015) on Penn Treebank (Marcus et al., 1993) trained by various optimizers in Table 11, indicating the consistent improvements by SEMA. Then, we evaluate fine-tuning with pre-trained BERT-Base (Devlin et al., 2018) backbone for text classification on Yelp Review (Yel) using USB settings (Wang et al., 2022) and language modeling with randomly initialized BERT-Base on WikiText-103 (Ott et al., 2019) uses FlowFormer settings. Table 12 shows that applying SEMA to pre-training or fine-tuning is more efficient than other WA methods. View Appendix B.7 for detailed settings.

## 4.4 Ablation Study

This section analyzes the two hyperparameters $\alpha$ and $T$ in SEMA to verify whether their default values are robust and general enough based on experimental settings in Sec. 4.1.

**Switching Interval $T$.** We first verify whether the one-epoch switching interval is optimal and robust for general usage. Table 13 shows that one epoch switching interval yields the optimal performance in most cases. However, choosing a smaller interval

Table 13: Ablation of switching interval (0.5∼5 epochs).

| $T$ | CIFAR-100 | STL-10 | IN-1K | CIFAR-10 | AgeDB | Yelp |
|---|---|---|---|---|---|---|
| Task | Cls (Acc)↑ | CL (Acc)↑ | MIM (Acc7↑ | Cls (Acc)↑ | Gen (FID)↓ | Reg (MAE)↓ | Cls (Acc)↑ |
| | WRN-28-10 | MoCo.V3 | SimMIM | DeiT-S | DDPM | R-50 | BERT |
| 0.5 | 82.23 | 50.73 | 92.01 | 79.5 | 6.07 | 7.34 | 68.17 |
| 1 | **82.35** | **52.27** | **92.06** | **80.6** | 5.30 | **7.22** | **68.46** |
| 2 | 82.34 | 52.25 | 91.93 | 80.1 | 5.33 | 7.24 | 68.28 |
| 5 | 82.08 | 51.94 | 91.68 | 78.9 | **5.28** | 7.26 | 68.23 |

hinders accurate gradient estimation and might disrupt the continuity of optimizer statistics in the Adam series and degenerate performance. On the contrary, larger intervals lead to slower update

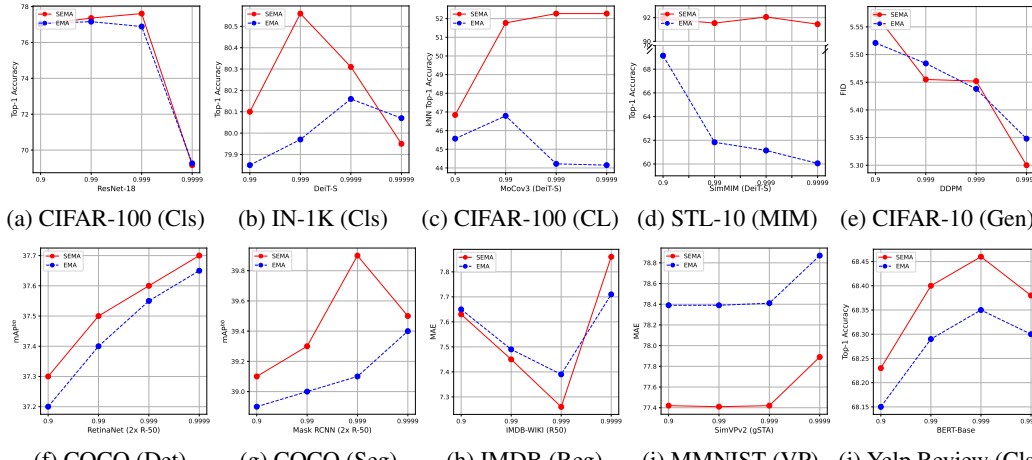

(a) CIFAR-100 (Cls)    (b) IN-1K (Cls)    (c) CIFAR-100 (CL)    (d) STL-10 (MIM)    (e) CIFAR-10 (Gen)

(f) COCO (Det)    (g) COCO (Seg)    (h) IMDB (Reg)    (i) MMNIST (VP)    (j) Yelp Review (Cls)

Figure 5: Ablation of the momentum $\alpha$ in EMA and SEMA, searching in the range of 0.9~0.9999 and 0.99~0.99999. SEMA shows robust choices of $\alpha$ across the most tasks.

rates and increased training time, except for specific tasks like diffusion generation that extremely prefer smoothness (Karras et al., 2023).

**Momentum Coefficient** $\alpha$. To investigate the effectiveness and generalization of SEMA, we conducted ablation studies with different momentum coefficients on SEMA and EMA, varying from 0.9 to 0.99999 (choosing four typical values). As shown in Figure 5, SEMA prefers 0.999 in most cases, except for image generation (0.9999) and video prediction (0.9). The preference of $\alpha$ for both EMA and SEMA are robust, and full values in different tasks are provided in Table A1 and Table A2.

## 5 CONCLUSION

This paper presents SEMA, a highly effective regularizer for DNN optimization that harmoniously blends the benefits of flatness and sharpness. SEMA has shown superior performance gains and versatility across various tasks, including discriminative and generative foundational tasks, regression, forecasting, and two modalities. As a pluggable and general method, SEMA expedites convergence and enhances final performances without incurring extra computational costs. SEMA marks a significant milestone in DNN optimization, providing a universally applicable solution for a multitude of deep learning training.

**Limitations and Future Works** SEMA achieves a delicate balance among several desirable attributes: it introduces no additional computational overhead, maintains user-friendliness, delivers performance gains, possesses plug-and-play capability, and demonstrates universal generalization. Consequently, it emerges as an ideal "free-lunch" optimization technique. The potential drawback, albeit minor, is that it may yield slightly smaller performance gains in certain scenarios. However, as a novel regularization technique, SEMA still harbors untapped potential. Notably, it is envisioned that future enhancements will enable more flexible, cost-free switching operations, allowing for adaptive adjustments of the switching interval. This adaptive capability would further unlock SEMA's latent potential, facilitating better performance optimization across diverse applications.

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

## A  PROOF OF PROPOSITION

Taking SGD as an example, we provide two propositions and their proofs of SEMA to investigate the favorable properties mentioned in Sec. 3.2 and Sec. 3.3. Let $\eta$ be the learning rate of the optimizer and $\lambda$ be the decay rate of SGD, EMA, and SEMA. In the noise quadratic model, the loss of iterates $\Theta_t$ is defined as:

$$\mathcal{L}(\Theta_t) = \frac{1}{2}(\Theta_t - c)^T A (\Theta_t - c),$$

where $c \sim \mathcal{N}(\theta^*, \Sigma)$. Without loss of generality, we set $\theta^* = 0$. Then, we can define the iterates of SGD, EMA, and SEMA (denoted as $\theta_t$, $\tilde{\theta}_t$ and $\theta_t^*$ respectively) as follows:

$$\begin{aligned}
\theta_t &= \theta_{t-1} - \eta \nabla \mathcal{L}(\theta_{\sqcup - \infty}), \\
\tilde{\theta}_t &= \lambda \theta_t + (1 - \lambda)\tilde{\theta}_{t-1}, \\
\theta_t' &= \theta_{t-1}^* - \eta \nabla \mathcal{L}(\theta_{\sqcup - \infty}^*), \\
\theta_t^* &= \lambda \theta_t' + (1 - \lambda)\theta_{t-1}^*, \\
\theta_0 &= \tilde{\theta}_0 = \theta_0^*.
\end{aligned}$$

Notice that iterates of $\theta_t$ and $\tilde{\theta}_t$ are defined jointly, and $\theta_t'$ is an intermediate iterate assisting to define the iterate of $\theta_t^*$.

### A.1  PROOF OF PROPOSITION 1

**Proposition 1 (Low-frequency Oscillation).**  In the noisy quadratic model, the variance of SGD, EMA, and SEMA iterates, denoted as $V_{\text{SGD}}^{(t)} := \mathbb{V}(\Theta_t^{\text{SGD}})$, $V_{\text{EMA}}^{(t)} := \mathbb{V}(\Theta_t^{\text{EMA}})$, and $V_{\text{SEMA}}^{(t)} := \mathbb{V}(\Theta_t^{\text{SEMA}})$ respectively, converge to following fixed points, *i.e.*, $V_{\text{SEMA}} < V_{\text{EMA}} < V_{\text{SGD}}$:

$$\begin{aligned}
V_{\text{SGD}} &= \frac{\eta A}{2I - \eta A}\Sigma, \\
V_{\text{EMA}} &= \frac{\lambda}{2 - \lambda} \cdot \frac{2 - \lambda - (1 - \lambda)\eta A}{\lambda + (1 - \lambda)\eta A} \cdot V_{\text{SGD}}, \\
V_{\text{SEMA}} &= \frac{2 - \lambda}{\lambda} \cdot \frac{\lambda + (1 - \lambda)\eta A}{2 - \lambda - (1 - \lambda)\eta A} \cdot \frac{\lambda \eta A}{2I - \lambda \eta A} \cdot \frac{2I - \eta A}{\eta A} \cdot V_{\text{EMA}}.
\end{aligned}$$

*Proof.*  First, we compute the stochastic dynamics of SGD, EMA, and SEMA. According to the property of variance and quadratic loss, we can get stochastic variance dynamics from iterates of trajectories:

$$\begin{aligned}
V(\theta_t) &= (I - \eta A)^2 V(\theta_{t-1}) + \eta^2 A^2 \Sigma, \\
V(\tilde{\theta}_t) &= \lambda^2 V(\theta_t) + (1 - \lambda)^2 V(\tilde{\theta}_{t-1}) + 2\lambda(1 - \lambda)Cov(\theta_t, \tilde{\theta}_{t-1}), \\
Cov(\theta_t, \tilde{\theta}_{t-1}) &= \lambda(I - \eta A)V(\theta_{t-1}) + (1 - \lambda)(I - \eta A)Cov(\theta_{t-1}, \tilde{\theta}_{t-2}), \\
V(\theta_t^*) &= (I - \lambda \eta A)^2 V(\theta_{t-1}^*) + \lambda^2 \eta^2 A^2 \Sigma.
\end{aligned}$$

Then, using Banachs fixed point theorem, we can easily derive $V_{\text{SGD}}$, $V_{\text{EMA}}$ and $V_{\text{SEMA}}$ as follows:

$$\begin{aligned}
V_{\text{SGD}} &= (I - \eta A)^2 V_{\text{SGD}} + \eta^2 A^2 \Sigma, \\
V_{\text{SGD}} &= \frac{\eta A}{2I - \eta A}\Sigma, \\
V_{\text{EMA}} &= \lambda^2 V_{\text{SGD}} + (1 - \lambda)^2 V_{\text{EMA}} + 2\lambda(1 - \lambda)\text{Cov}(\text{SGD, EMA}), \\
\text{Cov}(\text{SGD, EMA}) &= \lambda(I - \eta A)V_{\text{SGD}} + (1 - \lambda)(I - \eta A)\text{Cov}(\text{SGD, EMA}), \\
V_{\text{EMA}} &= \frac{\lambda}{2 - \lambda} \cdot \frac{2 - \lambda - (1 - \lambda)\eta A}{\lambda + (1 - \lambda)\eta A} \cdot V_{\text{SGD}}, \\
V_{\text{SEMA}} &= (I - \lambda \eta A)^2 V_{\text{SEMA}} + \lambda^2 \eta^2 A^2 \Sigma, \\
V_{\text{SEMA}} &= \frac{\lambda \eta A}{2I - \lambda \eta A}\Sigma.
\end{aligned}$$

Finally, we will prove inequality $V_{\text{SEMA}} < V_{\text{EMA}} < V_{\text{SGD}}$. To prove this, we only need to show the following two coefficients less or equal to one:

$$\text{coef}_1 = \frac{\lambda}{2-\lambda} \cdot \frac{2-\lambda-(1-\lambda)\eta A}{\lambda+(1-\lambda)\eta A},$$

$$\text{coef}_2 = \frac{2-\lambda}{\lambda} \cdot \frac{\lambda+(1-\lambda)\eta A}{2-\lambda-(1-\lambda)\eta A} \cdot \frac{\lambda\eta A}{2I-\lambda\eta A} \cdot \frac{2I-\eta A}{\eta A}.$$

For $\text{coef}_1$, we can see it is a decreasing function w.r.t $\eta$, and $\text{coef}_1 = 1$ when $\eta = 0$. Thus for $\eta > 0$, $\text{coef}_1 < 1$. For $\text{coef}_2$, we can simplify it as following:

$$\text{coef}_2 = \frac{\lambda+(1-\lambda)\eta A}{1-\frac{1-\lambda}{2-\lambda}\eta A} \cdot \frac{2I-\eta A}{2I-\lambda\eta A}.$$

Because learning rate $\eta$ and decay rate $\lambda$ are both quite small numbers, one can easily show that these two terms are smaller than 1, thus $\text{coef}_2 < 1$.

## A.2   PROOF OF PROPOSITION 2

**Proposition 2 (Fast Convergence).**   The iteration of SEMA ensures the gradient descent property, which SGD has but EMA doesn't have. More specifically expressed as:

$$(\theta_{t+1}^* - \theta_t^*) \propto -\nabla\mathcal{L}(\theta_{\sqcup}).$$

*Proof.*   This property is easily obtained by putting $\theta_t' = \theta_{t-1}^* - \eta\nabla\mathcal{L}(\theta_{\sqcup-\infty}^*)$ into $\theta_t^* = \lambda\theta_t' + (1-\lambda)\theta_{t-1}^*$, then we have:

$$\theta_t^* = \theta_{t-1}^* - \lambda\eta\nabla\mathcal{L}(\theta_{\sqcup-\infty}^*).$$

Let $\mathcal{L}(\theta_{\sqcup})$ denote the loss function evaluated at the parameters $\theta_t$ at time $t$, and $\mathcal{L}(\theta_{\sqcup-\infty}^*)$ denote the loss function evaluated at the exponentially moving average (EMA) parameters $\theta_{t-1}^*$ at time $t-1$.

**Integrated Analysis.**   As substantiated by *Propositions* above and the evidence in Figure 3, SEMA converges significantly faster to a local optimum than EMA does because of its effective amalgamation of the gradient descent characteristics of the baseline model and the stability advantage of EMA. The optimization process of SEMA will efficiently steer towards local minima unimpeded by slow or irregular parameter updates. In contrast, EMA lacks this benefit, potentially leading to its ensnaring in a flat but inferior local basin.

## A.3   PROOF OF PROPOSITION 3

**Proposition 3 (Superior Error Bound).**   Building on assumptions of a fixed learning rate and convergence properties of SGD in (Bottou et al., 2016), the error bound of SGD is $\mathcal{E}_{\text{SGD}} := \mathbb{E}[\mathcal{L}(\theta^{\text{EMA}}) - \mathcal{L}(\theta^*)]$, and error bounds for SGD, EMA, and SEMA can be ranked as, $\mathcal{E}_{\text{SEMA}} < \mathcal{E}_{\text{EMA}} < \mathcal{E}_{\text{SGD}}$:

$$\mathcal{E}_{\text{SGD}} \leq \frac{\eta LM}{2C\mu},$$

$$\mathcal{E}_{\text{EMA}} \leq \frac{(1-\alpha)\eta LM}{2C\mu},$$

$$\mathcal{E}_{\text{SEMA}} \leq \frac{\eta LM}{2\sigma_T C\mu},$$

$$\sigma_T \geq \frac{\mathbb{E}[\mathcal{L}(\theta_T^{\text{EMA}})] - \mathbb{E}[\mathcal{L}(\theta_{2T}^{\text{SEMA}})]}{\mathbb{E}[\mathcal{L}(\theta_T^{\text{EMA}})] - \mathbb{E}[\mathcal{L}(\theta_{2T}^{\text{EMA}})]},$$

where $L$ and $C > 0$ are the Lipschitz of $L(\mathcal{L})$ and its constant, $\mu > 1$ and $M > 1$ are the coefficients, and $\sigma_T$ denotes the error-bound improvement by switching once at the $T$ iteration.

*Proof.*   This property can be proofed with three steps based on assumptions from our previous properties and (Bottou et al., 2016).

*Step 1: the Error bound of SGD.* The objective function $\mathcal{L}$ and the SG satisfy the following assumptions: (a) The sequence of iterates $\theta_p$ is contained in an open set over which $\mathcal{L}$ is bounded below by a scalar $\mathcal{L}_{\inf}$. (b) There exist scalars $\mu_G \geq \mu > 0$ such that, for all $p \in \mathbb{N}$,

$$\nabla\mathcal{L}(\theta_p)^T \mathbb{E}\xi p[g(\theta_p, \xi_p)] \geq \mu|\nabla\mathcal{L}(\theta_p)|2^2 \quad \text{and} \quad |\mathbb{E}\xi_p[g(\theta_p, \xi_p)]|2 \leq \mu G|\nabla\mathcal{L}(\theta_p)|_2.$$

The function $g$ herein represents the classical method of Robbins and Monro, or it may alternatively signify a stochastic Newton or quasi-Newton direction. (c) There exist scalars $M \geq 0$ and $M_V \geq 0$ such that, for all $p \in \mathbb{N}$,

$$\mathbb{V}\xi p[g(\theta_p, \xi_p)] \leq M + M_V|\nabla\mathcal{L}(\theta_p)|_2^2$$

The first condition merely requires the objective function to be bounded below the region explored by the algorithm. The second requirement states that, in expectation, the vector $-g(\theta_p, \xi_p)$ is a direction of sufficient descent for $\mathcal{L}$ from $\theta_p$ with a norm comparable to the norm of the gradient. The third requirement states that the variance of $g(\theta_p, \xi_p)$ is restricted but in a relatively minor manner. The objective function $\mathcal{L} : \mathbb{R}^d \to \mathbb{R}$ is strongly convex in that there exists a constant $C > 0$ such that

$$\mathcal{L}(\overline{\theta}) \geq \mathcal{L}(\theta) + \nabla\mathcal{L}(\theta)^T(\overline{\theta} - \theta) + \frac{1}{2}C|\overline{\theta} - \theta|2^2 \quad \text{for all } (\overline{\theta}, \theta) \in \mathbb{R}^d \times \mathbb{R}^d.$$

Hence, $\mathcal{L}$ has a unique minimizer, denoted as $\theta^* \in \mathbb{R}^d$ with $\mathcal{L} := \mathcal{L}(\theta)$. For a strongly convex objective with a fixed stepsize, suppose that the SG method is run with a fixed stepsize, $\alpha_p = \overline{\alpha}$ *for all* $p \in \mathbb{N}$, satisfying:

$$0 < \overline{\alpha} \leq \frac{\mu}{\mu_G L}.$$

Then, the expected optimality gap satisfies the following inequality for all $p \in \mathbb{N}$:

$$\mathbb{E}[\mathcal{L}(\theta_p) - \mathcal{L}*] \leq \frac{\overline{\alpha}LM}{2C\mu} + (1 - \overline{\alpha}C\mu)^{p-1}\left(\mathcal{L}(\theta 1) - \mathcal{L} - \frac{\overline{\alpha}LM}{2C\mu}\right) \xrightarrow{p\to\infty} \frac{\overline{\alpha}LM}{2C\mu}.$$

Therefore, the error bound for SGD can be concisely represented as $\mathcal{E}\text{SGD} := \frac{\eta LM}{2C\mu}$, where $\eta = \overline{\alpha}$ is the fixed step size. The error bound suggests that SGD can converge relatively quickly, especially when the objective function $\mathcal{L}$ is strongly convex (indicated by a large value of $C$), the stochastic gradients $g(\theta_p, \xi_p)$ have a small variance (small $M$), and the sufficient descent condition is well-satisfied (large $\mu$). Additionally, a smaller step size $\eta$ can lead to a tighter error bound, although excessively small step sizes may result in slow convergence.

*Step 2: the Error bound of EMA.* Suppose the stochastic gradient (SG) method is executed with a stepsize sequence $\alpha_p$ such that, for all $p \in \mathbb{N}$,

$$\alpha_p = \frac{\beta}{\gamma + p}, \quad \beta > \frac{1}{C\mu}, \quad \gamma > 0, \quad \alpha_1 \leq \frac{\mu}{\mu_G}$$

where $C$ is the strong convexity constant of the objective function $\mathcal{L}$, and $\mu$ and $\mu_G$ are constants derived from the assumption (b) on the stochastic gradient $g(\theta_p, \xi_p)$. From the previous error-bound analysis for SGD, we have the following:

$$\mathbb{E}[\mathcal{L}(\theta_p) - \mathcal{L}] \leq \frac{\alpha_p LM}{2C\mu} + (1 - \alpha_p C\mu)^{p-1}(\mathcal{L}(\theta_1) - \mathcal{L} - \frac{\alpha_p LM}{2C\mu})$$

Substituting $\alpha_p$ and applying the geometric series formula, we obtain:

$$\mathbb{E}[\mathcal{L}(\theta_p) - \mathcal{L}^*] \leq \frac{\beta LM}{2C\mu(\gamma + p)} + \left(1 - \frac{\beta C\mu}{\gamma + p}\right)^{p-1}\left(\mathcal{L}(\theta_1) - \mathcal{L}^* - \frac{\beta LM}{2C\mu(\gamma + 1)}\right)$$

$$\leq \frac{\beta LM}{2C\mu(\gamma + p)} + \left(\frac{\gamma}{\gamma + p}\right)^{p-1}\left(\mathcal{L}(\theta_1) - \mathcal{L}^* - \frac{\beta LM}{2C\mu(\gamma + 1)}\right)$$

$$\leq \frac{\beta^2 LM}{2C\mu(\gamma + p)} + \frac{\gamma^p}{\gamma + p}\left(\mathcal{L}(\theta_1) - \mathcal{L}^*\right)$$

Since $\frac{\gamma^p}{\gamma+p} \leq 1$ and $\beta > \frac{1}{C\mu}$, as $p \to \infty$, the upper bound converges to:

$$\mathbb{E}[\mathcal{L}(\theta_p) - \mathcal{L}] \leq \frac{\beta^2 LM}{2C\mu(\gamma+p)}$$

Therefore, we can concisely represent the error bound for EMA as:

$$\mathcal{E}_{\text{EMA}} := \frac{\beta LM}{2C\mu(\gamma+p)}$$

It is noteworthy that since $\alpha_p$ is a decreasing sequence, $\alpha_p \to 0 \quad (p \to \infty)$, EMA can be viewed as a form of exponential decay of the learning rate for the SGD algorithm:

$$\alpha_p = \frac{\beta}{\gamma+p} \quad 1 - \alpha_p = 1 - \frac{\beta}{\gamma+p} = \frac{\gamma}{\gamma+p}$$

This indicates that the EMA decay factor $(1 - \alpha_p)$ directly influences the learning rate decay behavior.

Consequently, we can simplify the description of the EMA error bound as $\mathcal{E}_{\text{EMA}} := \frac{(1-\alpha_p)\eta LM}{2C\mu}$, where $\eta$ is the initial learning rate. Compared to the standard SGD, the EMA error bound $\mathcal{E}_{\text{EMA}}$ is larger, indicating a slower convergence rate. This is because the term $(1 - \alpha_p)$ approaches 1 as $p \to \infty$, resulting in a slower decay of the error bound. While the decaying learning rate in the later iterations assists EMA in converging to a local minimum by mitigating oscillations caused by large step sizes, this approach is not without drawbacks. One significant issue is the accumulation of bias, which arises from the EMA of the gradients. As the iterations progress, the gradients from earlier steps contribute less and less to the update, leading to a bias towards more recent gradients. Furthermore, in non-smooth settings, the EMA of momentum can actually impair the theoretical worst-case convergence rate. The smoothing effect introduced by EMA can hinder the algorithm's ability to navigate through non-differentiable regions of the objective landscape, potentially slowing down convergence or even causing the algorithm to converge to suboptimal solutions.

*Step 3: the Error bound of SEMA.* Based on the error bound of EMA, the advantage of SEMA stems from the switching mechanism every $T$ iteration, which can be formulated as comparing the reduction of errors between using switching or not at the $t$-th iteration and the $(t + T)$-th iteration,

$$\sigma_T = \sum_{l=1}^{\lfloor \frac{t}{T} \rfloor} \frac{\mathbb{E}[\mathcal{L}(\theta_{lT}^{\text{EMA}})] - \mathbb{E}[\mathcal{L}(\theta_{(l+1)T}^{\text{SEMA}})]}{\mathbb{E}[\mathcal{L}(\theta_{lT}^{\text{EMA}})] - \mathbb{E}[\mathcal{L}(\theta_{(l+1)T}^{\text{EMA}})]},$$

where $l$ is the total switching time during training, $\lfloor \frac{t}{T} \rfloor > l > 0$. To simplify the proof, the lower bound of the gains from switching can be calculated,

$$\sigma_T \geq \frac{\mathbb{E}[\mathcal{L}(\theta_T^{\text{EMA}})] - \mathbb{E}[\mathcal{L}(\theta_{2T}^{\text{SEMA}})]}{\mathbb{E}[\mathcal{L}(\theta_T^{\text{EMA}})] - \mathbb{E}[\mathcal{L}(\theta_{2T}^{\text{EMA}})]}.$$

The numerator, $\mathbb{E}[\mathcal{L}(\theta_T^{\text{EMA}})] - \mathbb{E}[\mathcal{L}(\theta_{2T}^{\text{SEMA}})]$, accumulates over the interval from $T$ to $2T$ and experiences a switch at time $2T$, mirroring the update of $\mathbb{E}[\mathcal{L}(\theta_{2T}^{\text{SGD}})]$. Conversely, the denominator reflects the update of $\mathbb{E}[\mathcal{L}(\theta_T^{\text{EMA}})]$ from $T$ to $2T$ in the absence of a switch, yielding $\mathbb{E}[\mathcal{L}(\theta_{2T}^{\text{EMA}})]$. Leveraging the fact that the error bound of EMA is superior to that of SGD, we infer that $\mathbb{E}[\mathcal{L}(\theta_{2T}^{\text{SEMA}})] \leq \mathbb{E}[\mathcal{L}(\theta_{2T}^{\text{SGD}})] + (\mathbb{E}[\mathcal{L}(\theta_T^{\text{EMA}})] - \mathbb{E}[\mathcal{L}(\theta_{2T}^{\text{EMA}})])$ and $\sigma_T > 1$. Therefore, we have

$$\mathcal{E}_{\text{SEMA}} = \frac{(1-\alpha_T)\eta LM}{2C\mu} \cdot \frac{1}{\sigma_T} < \mathcal{E}_{\text{EMA}}.$$

Our analysis of $\sigma_T$ unveils a pivotal aspect of the convergence dynamics of SEMA. As $\sigma_T$ accumulates across successive iterations, it signifies a gradual diminution in the discrepancy between the expected loss values under switching and non-switching conditions. This discernment implies that as $\sigma_T$ approaches unity with a decreasing trend, the upper bound on SEMA's error constricts progressively, yielding a tighter error bound and faster convergence compared to EMA. Furthermore, the switching mechanism in SEMA effectively mitigates the accumulation of bias inherent to EMA, thereby enhancing the overall convergence behavior and optimality of the solution. In summary,

from the derived error bounds, we can conclude that SEMA has the smallest error bound among SGD, EMA, and SEMA, satisfying:

$$\mathcal{E}_{\text{SEMA}} < \mathcal{E}_{\text{SGD}} < \mathcal{E}_{\text{EMA}}$$

This indicates that SEMA not only converges faster than EMA but also exhibits a tighter error bound compared to the standard SGD algorithm, making it a more effective optimization method for strongly convex objectives. While the error-bound analysis assumes strong convexity, SEMA's ability to mitigate bias accumulation and adapt its convergence behavior through the switching mechanism can prove advantageous in the highly non-linear and non-convex settings characteristic of DNN optimization.

## A.4   SHARPNESS VS. FLATNESS

**Sharpness**: Sharpness refers to the depth of the local minima. Specifically, it measures how steep the loss function is around the local minimum. Mathematically, sharpness can be quantified by the eigenvalues of the Hessian matrix of the loss function at the local minimum. A deeper minimum (higher sharpness) corresponds to larger eigenvalues of the Hessian, indicating a more pronounced curvature in the loss landscape (Foret et al., 2021; Kaddour et al., 2022).

Formally, given a local minimum $\theta^*$ of the loss function $\mathcal{L}(\theta)$, the sharpness $S(\theta^*)$ can be defined as:

$$S(\theta^*) = \max_{\theta \in N(\theta^*)} \lambda_{\max}(\nabla^2 L(\theta)),$$

where $\mathcal{N}(\theta^*)$ is a neighborhood around $\theta^*$, and $\nabla^2 \mathcal{L}(\theta)$ denotes the Hessian matrix of $\mathcal{L}$ at $\theta$. Here, $\max \lambda_{\max}(\nabla^2 L(\theta))$ refers to the maximum eigenvalues of the Hessian matrix.

The Sharpness (Extreme) represents a very sharp loss landscape, where the loss function changes rapidly with small variations in parameters. This extreme sharpness, while potentially leading to a deep minimum, can result in poor generalization.

**Flatness**: Flatness in the context of loss landscapes refers to the width of the local minima. It measures how wide the basin of attraction is around the local minimum. A flatter minimum (higher flatness) corresponds to smaller eigenvalues of the Hessian matrix of the loss function at the local minimum, indicating a broader and less steep region around the minimum (Li et al., 2017; Keskar et al., 2016).

Formally, the flatness $F(\theta^*)$ can be defined as:

$$F(\theta^*) = \min_{\theta \in N(\theta^*)} \lambda_{\min}(\nabla^2 L(\theta)),$$

where $\mathcal{N}(\theta^*)$ is a neighborhood around $\theta^*$, and $\nabla^2 \mathcal{L}(\theta)$ denotes the Hessian matrix of $\mathcal{L}$ at $\theta$. Here, $\min \lambda_{\min}(\nabla^2 L(\theta))$ refers to the minimum eigenvalues of the Hessian matrix.

The Flatness represents a very flat loss landscape, where the loss function is relatively constant over a wide range of parameters. This extreme flatness, while stable, may not necessarily lead to optimal generalization.

**Trade-off Between Flatness and Sharpness:** The relative scale between sharpness and flatness is crucial for characterizing the loss landscape. In our paper, we consider the ratio of the maximum to the minimum eigenvalues of the Hessian matrix at the local minimum. This ratio provides a measure of the anisotropy of the loss landscape, which is essential for understanding the trade-off between sharpness and flatness. Formally, the ratio $R(\theta^*)$ can be defined as:

$$R(\theta^*) = \frac{\max_{\theta \in \mathcal{N}(\theta^*)} \left(\nabla^2 \mathcal{L}(\theta)\right)}{\min_{\theta \in \mathcal{N}(\theta^*)} \left(\nabla^2 \mathcal{L}(\theta)\right)}.$$

A higher or lower ratio is generally unfavorable for generalization, as it indicates an imbalance between sharpness and flatness. A balanced ratio, neither too high nor too low, indicates a more optimal trade-off between the depth and width of the local minimum, which we interpret as a balanced trade-off between sharpness and flatness. *SEMA's Role in Achieving Optimal Trade-off*: SEMA is designed to balance the trade-off between flatness and sharpness dynamically. By switching the

EMA parameters to the original model after each epoch, SEMA leverages the fast convergence of EMA to reach flat local minima while allowing the optimizer to explore sharper trajectories based on previous EMA parameters. This dynamic regularization helps DNNs to reach generalization optima that better trade-off between flatness and sharpness, leading to improved performance and faster convergence.

This figure A1 provides a detailed visualization of the loss landscapes for different optimization methods, highlighting the critical trade-off between sharpness and flatness. Sharpness refers to the depth of the local minima, quantified by the eigenvalues of the Hessian matrix of the loss function at the local minimum. A deeper minimum (higher sharpness) corresponds to larger eigenvalues, indicating a more pronounced curvature in the loss landscape. The baseline model, using standard optimizers without EMA, exhibits steep peaks indicative of high sharpness, represented by the blue line, which may lead to unstable convergence and suboptimal gener-

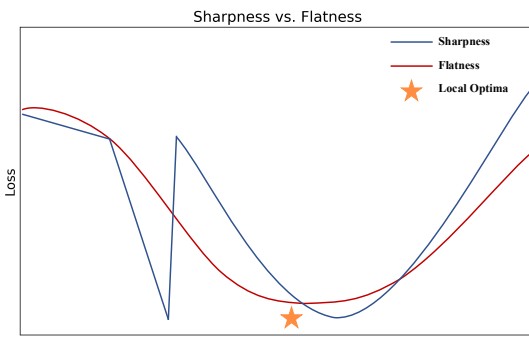

Figure A1: Illustration of the trade-off between extreme flatness and sharpness in loss landscapes.

alization. In contrast, flatness refers to the width of the local minima, quantified by smaller eigenvalues of the Hessian matrix, indicating a broader and less steep region around the minimum. The EMA model demonstrates smoother curves with lower peaks, signifying enhanced flatness, represented by the red line, and more stable minima, though it may overlook deeper optima. The SEMA model, by dynamically switching between EMA and the original model, effectively balances both sharpness and flatness. It achieves deeper and smoother local minima, allowing the optimizer to explore lower basins while maintaining stability. This balanced approach results in superior performance and generalization across diverse applications, as evidenced by smoother decision boundaries and accelerated convergence speeds.

## B  IMPLEMENTATION DETAILS

This section provides implementation settings and dataset information for empirical experiments conducted in Sec. 4. We follow existing benchmarks for all experiments to ensure fair comparisons. As for the compared WA methods and Lookahead, we adopt the following settings: EMA is applied as test-time regularization during the whole training process with tuned momentum coefficient; SWA applies the 1.25 budget (*e.g.,* ensemble five models iteratively); Lookahead applies the slow model learning rate with $\alpha = 0.5$ and the update interval $k = 100$. Meanwhile, SEMA uses "by-epoch" switching, (*i.e., T* is the iteration number of traversing the entire training set once, and the momentum ratios for different tasks are shown in Table A1 and Table A2.

Table A1: Momentum coefficient $\alpha$ in EMA and SEMA for vision applications, including image classification (Cls.), self-supervised learning (SSL) with contrastive learning (CL) methods or masked image modeling (MIM) methods, objection detection (Det.) and instance segmentation (Seg.), and image generation (Gen.) tasks.

| Dataset | CIFAR-100 | | | STL-10 | | ImageNet-1K | | | CIFAR-10 | CelebA | COCO | |
| --- | --- | --- | --- | --- | --- | --- | --- | --- | --- | --- | --- | --- |
| Task | Cls. | SSL (CL) | SSL (MIM) | SSL (CL) | SSL (MIM) | Cls. | SSL (CL) | SSL (MIM) | Gen. | Gen. | Det. | Seg. |
| EMA | 0.99 | 0.999 | 0.9 | 0.9999 | 0.999 | 0.9999 | 0.99996 | 0.999 | 0.9999 | 0.9999 | 0.9999 | 0.9999 |
| SEMA | 0.99 | 0.999 | 0.999 | 0.999 | 0.999 | 0.999 | 0.9999 | 0.999 | 0.9999 | 0.9999 | 0.999 | 0.999 |

Table A2: Momentum coefficient $\alpha$ in EMA and SEMA for regression (Reg.), video prediction (VP), language processing (LP), text classification (Cls.), and language modeling (LM) tasks.

| Dataset | RCFMNIST | AgeDB | IMDB-WIKI | MMNIST | Penn TreeBank | Yelp Review | WikiText-103 |
| --- | --- | --- | --- | --- | --- | --- | --- |
| Task | Reg. | Reg. | Reg. | VP | LP | Cls. | LM |
| EMA | 0.999 | 0.999 | 0.999 | 0.999 | 0.9999 | 0.9999 | 0.9999 |
| SEMA | 0.999 | 0.999 | 0.999 | 0.999 | 0.999 | 0.999 | 0.999 |

Table A3: Ingredients and hyper-parameters used for ImageNet-1K training settings with various optimizers. Note that the settings of PyTorch (Simonyan & Zisserman, 2014), RSB A2 and A3 (Wightman et al., 2021), and LARS (Ginsburg et al., 2018) take ResNet-50 as the examples, DeiT (Touvron et al., 2021) and Adan (Xie et al., 2023) settings take DeiT-S as the example, and the ConvNeXt (Liu et al., 2022b) setting is a variant of the DeiT setting for ConvNeXt and Swin Transformer.

| Procedure | PyTorch | DeiT | ConvNeXt | RSB A2 | RSB A3 | Adan | LARS |
| --- | --- | --- | --- | --- | --- | --- | --- |
| Train Resolution | 224 | 224 | 224 | 224 | 160 | 224 | 160 |
| Test Resolution | 224 | 224 | 224 | 224 | 224 | 224 | 224 |
| Test crop ratio | 0.875 | 0.875 | 0.875 | 0.95 | 0.95 | 0.85 | 0.95 |
| Epochs | 100 | 300 | 300 | 300 | 100 | 150 | 100 |
| Batch size | 256 | 1024 | 4096 | 2048 | 2048 | 2048 | 2048 |
| Optimizer | SGD | AdamW | AdamW | LAMB | LAMB | Adan | LARS |
| Learning rate | 0.1 | $1 \times 10^{-3}$ | $4 \times 10^{-3}$ | $5 \times 10^{-3}$ | $8 \times 10^{-3}$ | $1.6 \times 10^{-2}$ | $8 \times 10^{-3}$ |
| Optimizer Momentum | 0.9 | 0.9, 0.999 | 0.9, 0.999 | 0.9, 0.999 | 0.9, 0.999 | 0.98, 0.92, 0.99 | 0.9 |
| LR decay | Cosine | Cosine | Cosine | Cosine | Cosine | Cosine | Cosine |
| Weight decay | $10^{-4}$ | 0.05 | 0.05 | 0.02 | 0.02 | 0.02 | 0.02 |
| Warmup epochs | ✗ | 5 | 20 | 5 | 5 | 60 | 5 |
| Label smoothing $\epsilon$ | ✗ | 0.1 | 0.1 | ✗ | ✗ | 0.1 | ✗ |
| Dropout | ✗ | ✗ | ✗ | ✗ | ✗ | ✗ | ✗ |
| Stochastic Depth | ✗ | 0.1 | 0.1 | 0.05 | ✗ | 0.1 | ✗ |
| Repeated Augmentation | ✗ | ✓ | ✓ | ✓ | ✗ | ✗ | ✗ |
| Gradient Clip. | ✗ | 5.0 | ✗ | ✗ | ✗ | 5.0 | ✗ |
| Horizontal flip | ✓ | ✓ | ✓ | ✓ | ✓ | ✓ | ✓ |
| RandomResizedCrop | ✓ | ✓ | ✓ | ✓ | ✓ | ✓ | ✓ |
| Rand Augment | ✗ | 9/0.5 | 9/0.5 | 7/0.5 | 6/0.5 | 7/0.5 | 6/0.5 |
| Auto Augment | ✗ | ✗ | ✗ | ✗ | ✗ | ✗ | ✗ |
| Mixup $\alpha$ | ✗ | 0.8 | 0.8 | 0.1 | 0.1 | 0.8 | 0.1 |
| Cutmix $\alpha$ | ✗ | 1.0 | 1.0 | 1.0 | 1.0 | 1.0 | 1.0 |
| Erasing probability | ✗ | 0.25 | 0.25 | ✗ | ✗ | 0.25 | ✗ |
| ColorJitter | ✗ | ✗ | ✗ | ✗ | ✗ | ✗ | ✗ |
| EMA | ✗ | ✓ | ✓ | ✗ | ✗ | ✗ | ✗ |
| CE loss | ✓ | ✓ | ✓ | ✗ | ✗ | ✓ | ✗ |
| BCE loss | ✗ | ✗ | ✗ | ✓ | ✓ | ✗ | ✓ |

### B.1  IMAGE CLASSIFICATION

We evaluate popular weight average (WA) and optimization methods with image classification tasks based on various optimizers and network architectures on CIFAR-100 (Krizhevsky et al., 2009) and

Table A4: Ingredients and hyper-parameters used for pre-training on ImageNet-1K various SSL methods. Note that CL methods require complex augmentations proposed in SimCLR (SimCLR Aug.) and evaluated by Linear probing, while MIM methods use fine-tuning (FT) protocols.

| Configuration | SimCLR | MoCo.V2 | BYOL | Barlow Twins | MoCo.V3 | MAE | SimMIM/A$^2$MIM |
|---|---|---|---|---|---|---|---|
| Pre-training resolution | $224 \times 224$ | $224 \times 224$ | $224 \times 224$ | $224 \times 224$ | $224 \times 224$ | $224 \times 224$ | $224 \times 224$ |
| Encoder | ResNet | ResNet | ResNet | ResNet | ViT | ViT | ViT |
| Augmentations | SimCLR Aug. | SimCLR Aug. | SimCLR Aug. | SimCLR Aug. | SimCLR Aug. | RandomResizedCrop | RandomResizedCrop |
| Mask patch size | ✗ | ✗ | ✗ | ✗ | ✗ | $16 \times 16$ | $32 \times 32$ |
| Mask ratio | ✗ | ✗ | ✗ | ✗ | ✗ | 75% | 60% |
| Projector / Decoder | 2-MLP | 2-MLP | 2-MLP | 3-MLP | 2-MLP | ViT Decoder | FC |
| Optimizer | SGD | LARS | LARS | LARS | AdamW | AdamW | AdamW |
| Base learning rate | 4.8 | $3 \times 10^{-2}$ | 4.8 | 1.6 | $2.4 \times 10^{-3}$ | $1.2 \times 10^{-3}$ | $4 \times 10^{-4}$ |
| Weight decay | $1 \times 10^{-4}$ | $1 \times 10^{-6}$ | $1 \times 10^{-6}$ | $1 \times 10^{-6}$ | 0.1 | 0.05 | 0.05 |
| Optimizer momentum | 0.9 | 0.9 | 0.9 | 0.9 | 0.9, 0.95 | 0.9, 0.999 | 0.9, 0.999 |
| Batch size | 4096 | 256 | 4096 | 2048 | 4096 | 2048 | 2048 |
| Learning rate schedule | Cosine | Cosine | Cosine | Cosine | Cosine | Cosine | Step / Cosine |
| Warmup epochs | 10 | ✗ | 10 | 10 | 40 | 10 | 10 |
| Gradient Clipping | ✗ | ✗ | ✗ | ✗ | max norm= 5 | max norm= 5 | max norm= 5 |
| Evaluation | Linear | Linear | Linear | Linear | Linear | FT | FT |

ImageNet-1K (Deng et al., 2009). Experiments are implemented on `OpenMixup` (Li et al., 2022) codebase with 1 or 8 Nvidia A100 GPUs.

**ImageNet-1K.** We perform regular ImageNet-1K classification experiments following the widely used training settings with various optimizers and backbone architectures, as shown in Table A3. We consider popular backbone models, including ResNet (He et al., 2016), DeiT (Vision Transformer) (Dosovitskiy et al., 2020; Touvron et al., 2021), Swin Transformer (Liu et al., 2021), ConvNeXt (Liu et al., 2022b), and MogaNet (Li et al., 2024c). For all models, the default input image resolution is $224^2$ for training from scratch on 1.28M training images and testing on 50k validation images. ConvNeXt and Swin share the same training settings. As for augmentation and regularization techniques, we adopt most of the data augmentation and regularization strategies applied in DeiT training settings, including Random Resized Crop (RRC) and Horizontal flip (Szegedy et al., 2015), RandAugment (Cubuk et al., 2020), Mixup (Zhang et al., 2018), CutMix (Yun et al., 2019), random erasing (Zhong et al., 2020), ColorJitter (He et al., 2016), stochastic depth (Huang et al., 2016), and label smoothing (Szegedy et al., 2016). Note that EMA (Polyak & Juditsky, 1992) with the momentum coefficient of 0.9999 is basically adopted in DeiT and ConvNeXt training, but remove it as the baseline in Table 2. We also remove additional augmentation strategies (Cubuk et al., 2019; Liu et al., 2022d; Li et al., 2021; Liu et al., 2022c), *e.g.,* PCA lighting (Krizhevsky et al., 2012) and AutoAugment (Cubuk et al., 2019).

**CIFAR-100.** We use different training settings for a fair comparison of classical CNNs and modern Transformers on CIFAR-100, which contains 50k training images and 10k testing images of $32^2$ resolutions. As for classical CNNs with bottleneck structures, including ResNet variants, ResNeXt (Xie et al., 2017), Wide-ResNet (Zagoruyko & Komodakis, 2016), and DenseNet (Huang et al., 2017), we use $32^2$ resolutions with the CIFAR version of network architectures, *i.e.,* downsampling the input size to $\frac{1}{2}$ in the stem module instead of $\frac{1}{8}$ on ImageNet-1K. Meanwhile, we train three modern architectures for 200 epochs from the stretch. We resize the raw images to $224^2$ resolutions for DeiT-S and Swin-T while modifying the stem network as the CIFAR version of ResNet for ConvNeXt-T with $32^2$ resolutions.

B.2 SELF-SUPERVISED LEARNING

We consider two categories of popular self-supervised learning (SSL) on CIFAR-100, STL-10 (Coates et al., 2011), and ImageNet-1K: contrastive learning (CL) for discriminative representation and masked image modeling (MIM) for more generalizable representation. Experiments are implemented on `OpenMixup` codebase with 4 Tesla V100 GPUs.

**Contrastive Learning.** To verify the effectiveness of WA methods with CL methods with both self-teaching or non-teaching frameworks, we evaluate five classical CL methods on CIFAR-100, STL-10, and ImageNet-1K datasets, including SimCLR (Chen et al., 2020a), MoCo.V2 (Chen et al., 2020b), BYOL (Grill et al., 2020), Barlow Twins (Zbontar et al., 2021), and MoCo.V3 (Chen et al., 2020b). STL-10 is a widely used dataset for SSL or semi-supervised tasks, consisting of 5K labeled training images for 10 classes and 100K unlabelled training images, and a test set of 8K images in

$96^2$ resolutions. The pre-training settings are borrowed from their original papers on ImageNet-1K, as shown in Table A4. As for the SimCLR augmentations, the recipes include *RandomResizedCrop* with the scale in $[0.08, 1.0]$ and *RandomHorizontalFlip*, color augmentations of *ColorJitter* with {brightness, contrast, saturation, hue} strength of $\{0.4, 0.4, 0.4, 0.1\}$ with an applying probability of 0.8 and *RandomGrayscale* with an applying probability of 0.2, and blurring augmentations of a Gaussian kernel of size $23 \times 23$ with a standard deviation uniformly sampled in $[0.1, 2.0]$. We use the same setting on CIFAR-100 and STL-10, where the input resolution of CIFAR-100 is resized to $224^2$. The pre-training epoch on CIFAR-100 and STL-10 is 1000, while it is set to the official setup on ImageNet-1K, as shown in Table 5. As for the evaluation protocol, we follow MoCo.V2 and MoCo.V3 to conduct linear probing upon pre-trained representations. Note that MoCo.V2, BYOL, and MoCo.V3 require EMA with the momentum of 0.999, 0.99996, and 0.99996 as the teacher model, while SimCLR and Barlow Twins do not need WA methods by default.

**Masked Image Modeling.** As for generative pre-training with MIM methods (He et al., 2022; Li et al., 2023b), we choose SimMIM (Xie et al., 2022) and $A^2$MIM (Li et al., 2023a) to perform pre-training and fine-tuning with ViT (Dosovitskiy et al., 2020) on CIFAR-100, STL-10, and ImageNet-1K. Similarly, two MIM methods utilize their official pre-training setting on ImageNet-1K for three datasets, as shown in Table A4. CIFAR-100 and STL-10 use $224^2$ and $96^2$ resolutions to pre-train DeiT-S for 1000 epochs, while ImageNet-1K uses $224^2$ resolutions to pre-training ViT-B for 800 epochs. The fine-tuning evaluation protocols are also adopted as their original recipes, fine-tuning 100 epochs with a layer decay ratio of 0.65 with the AdamW optimizer. Note that both the MIM methods do not require WA techniques during pre-training.

### B.3 Object Detection and Instance Segmentation

Following Swin Transformers (Liu et al., 2021), we evaluate objection detection and instance segmentation tasks as the representative vision downstream tasks (Xiao et al., 2018; Li et al., 2020; 2024b) on COCO (Lin et al., 2014) dataset, which include 118K training images (*train2017*) and 5K validation images (*val2017*). Experiments of COCO detection and segmentations are implemented on `MMDetection` (Chen et al., 2019) codebase and run on 4 Tesla V100 GPUs.

**Fine-tuning.** Taking ImageNet pre-trained ResNet-50 and Swin-T as the backbone encoders, we adopt RetinaNet (Lin et al., 2017), Mask R-CNN (He et al., 2017), and Cascade Mask R-CNN (Cai & Vasconcelos, 2019) as the standard detectors. As for ResNet-50, we employ the SGD optimizer for training $2\times$ (24 epochs) and $3\times$ (36 epochs) settings with a basic learning rate of $2 \times 10^{-2}$, a batch size of 16, and a fixed step learning rate scheduler. Since MMDetection uses repeat augmentation for Cascade Mask R-CNN, its training $1\times$ and $3\times$ with multi-scale (MS) resolution and advanced data augmentations equals training $3\times$ and $9\times$, which can investigate the regularization capacities of WA techniques. As for Swin-T, we employ AdamW (Loshchilov & Hutter, 2019) optimizer for training $1\times$ schedulers (12 epochs) with a basic learning rate of $1 \times 10^{-4}$ and a batch size of 16. During training, the shorter side of training images is resized to 800 pixels, and the longer side is resized to not more than 1333 pixels. We calculate the FLOPs of compared models at $800 \times 1280$ resolutions. The momentum of EMA and SEMA is 0.9999 and 0.999.

**Training from Scratch.** For the YoloX (Ge et al., 2021) detector, we follow its training settings with randomly initialized YoloX-S encoders (the modified version of CSPDarkNet). The detector is trained 300 epochs by SGD optimizer with a basic learning rate of $1 \times 10^{-2}$, a batch size of 64, and a cosine annealing scheduler. During training, input images are resized to $640 \times 640$ resolutions and applied complex augmentations like Mosaic and Mixup (Zhang et al., 2018; Qin et al., 2024). The momentum of EMA and SEMA is 0.9999 and 0.999.

### B.4 Image Generation

Following DDPM (Ho et al., 2020), we evaluated image generation tasks based on CIFAR-10 (Krizhevsky et al., 2009), which includes 5K training images and 1K testing images (a total of 6K images). The training was conducted with a batch size of 128, performing 1000 training steps per second and using a learning rate of $2 \times 10^{-4}$. The DDPM-torch codebase was utilized to implement the DDPM image generation experiments executed on 4 Tesla V100 GPUs. Similarly, for

the CelebA dataset (Liu et al., 2015), which includes 162,770 training images, 19,867 validation images, and 19,962 testing images (a total of 202,599 images), image generation tasks were performed with a batch size of 128. The training was conducted with 1000 training steps per second, and the learning rate was set to 2e-5. The `ddpm-torch`[1] codebase was employed for implementing the DDPM image generation experiments, which were executed on 4 Tesla V100 GPUs. For the two datasets, the models utilized EMA and SEMA, with an ema decay factor set to the default value of 0.9999. The total number of epochs required for training the CIFAR-10 model is 2000 and 600 CelebA, optimized by Adam (Kingma & Ba, 2014) and a batch size of 128. In the final experiment, the model was trained once, and checkpoints were saved at regular intervals (every 50 epochs for CIFAR-10 and every 20 epochs for CelebA). Then, using 4 GPUs and the DDIM (Song et al., 2021) sampler, 50,000 samples were generated in parallel for each checkpoint. Finally, the FID score (Simonyan & Zisserman, 2014) was computed using the codebase to evaluate the 50,000 generated samples and record the results.

## B.5 VIDEO PREDICTION

Following SimVP (Gao et al., 2022) and OpenSTL (Tan et al., 2023), we verify WA methods with video prediction methods on Moving MNIST (Srivastava et al., 2015). We evaluate various Metaformer architectures (Yu et al., 2022) and MogaNet with video prediction tasks on Moving MNIST (MMNIST) (Lin et al., 2014) based on SimVP (Gao et al., 2022). Notice that the hidden translator of SimVP is a 2D network module to learn spatiotemporal representation, which any 2D architecture can replace. Therefore, we can benchmark various architectures based on the SimVP framework. In MMNIST (Srivastava et al., 2015), each video is randomly generated with 20 frames containing two digits in $64 \times 64$ resolutions, and the model takes 10 frames as the input to predict the next 10 frames. Video predictions are evaluated by Mean Square Error (MSE), Mean Absolute Error (MAE), and Structural Similarity Index (SSIM). All models are trained on MMNIST from scratch for 200 or 2000 epochs with Adam optimizer, a batch size of 16, a OneCycle learning rate scheduler, an initial learning rate selected in $\{1 \times 10^{-2}, 5 \times 10^{-3}, 1 \times 10^{-3}, 5 \times 10^{-4}\}$. Experiments of video prediction are implemented on `OpenSTL` codebase (Tan et al., 2023) and run on a single NVIDIA Tesla V100 GPU.

## B.6 VISUAL ATTRIBUTE REGRESSION

As for regression tasks, we conducted age regression experiments on two datasets: IMDB-WIKI (Rothe et al., 2018) and AgeDB (Moschoglou et al., 2017). AgeDB comprises images of various celebrities, encompassing actors, writers, scientists, and politicians, with annotations for identity, age, and gender attributes. The IMDB-WIKI dataset comprises approximately 167,562 face images, each associated with an age and gender label. The age range of the two datasets is from 1 to 101. In age regression tasks, our objective is to extract human features that enable the model to predict age as a continuous real value. Meanwhile, we also consider a rotation angle regression task on RCF-MNIST (Yao et al., 2022) dataset, which incorporates a more complex background inspired by CIFAR-10 to resemble natural images closely. This task allows the model to regress the rotation angle of the foreground object. We adopted the same experimental settings as described in SemiReward (Li et al., 2024a) and C-Mixup (Yao et al., 2022). In particular, we used ConvNeXt-T, ResNet-18, and ResNet-50 as backbone models, in addition to using a variety of methods for comparison. The input resolutions were set to $224^2$ for AgeDB and IMDB-WIKI and $32^2$ for RCF-MNIST. Models are optimized with $\ell_1$ loss and the AdamW optimizer for 400 or 800 epochs for AgeDB/IMDB-WIKI or RCF-MNIST datasets. MAE and RMSE are used as evaluation metrics.

## B.7 LANGUAGE PROCESSING

**Penn TreeBank with LSTM.** Following Adablief (Zhuang et al., 2020), the language processing experiment with LSTM (Ma et al., 2015) is conducted with Penn TreeBank dataset (Marcus et al., 1993) (with 887,521 training tokens, 70,390 validation tokens, 78,669 test tokens, vocab of 10,000, and 4.8% OoV) on LSTM with Adan as the baseline, utilizing its default weight decay (0.02) and $betas$ ($\beta_1 = 0.02$, $\beta_2 = 0.08$, $\beta_3 = 0.01$), and a learning rate of 0.01. We applied EMA and SEMA for comparison, and momentum defaults to 0.999 and 0.9999. We fully adhere to the experimental

---

[1]https://github.com/tqch/ddpm-torch/

settings in Adablief and use its codebase, applying the default settings for all other hyperparameters provided by Adablief (weight decay=1.2e-6, eps=1e-8, batch size=80, a total of 8000 epochs for single training). We use Perplexity as the primary evaluation metric for observing the training situation of SEMA.

**Text Classification with Yelp Review.** Second, for the Yelp review task in USB (Wang et al., 2022), we use the pre-trained BERT (Devlin et al., 2018) and experiment on the Yelp review dataset (Yel). The dataset contains a large number of user reviews and is made up of five classes (scores), each containing 130,000 training samples and 10,000 test samples. For BERT under USB, we adopt the Adam optimizer with a weight decay of 1e-4, a learning rate of 5e-5, and a layer decay ratio of 0.75, and the Momentum default values for SEMA and EMA are the same as before. We also fully adhere to and utilize its fully-supervised setting.

**Language Modeling with WikiText-103.** We also conducted language modeling experiments on WikiText-103 (Ott et al., 2019) (with 103,227,021 training tokens, 217,646 validation tokens, 245,569 test tokens, and a vocabulary of 267,735). In the comprehensive experimental, the sequence length was set to 512, and the experiment settings followed the specifications of fairseq (weight decay was 0.01, using the Adam optimizer, and learning rate of 0.0005). The default Momentum values for EMA and SEMA were maintained for training and evaluation, and the final comparison was based on Perplexity.

## C  EMPIRICAL EXPERIMENTS

**Loss Landscape.** The 1D linear interpolation method (Goodfellow et al., 2015) assesses the loss value along the direction between two minimizers of the same network loss function (Li et al., 2017). We visualize the 1d loss landscape 2b of the model guided by SEMA, using CNNs and Vits as backbones and SEMA showed sharper precision and loss lines compared to the baseline model, EMA and SAM, indicating better performance. To plot the 2d loss landscape 3, we select two random directions and normalize them in the same way as in the 1D plot. By observing the different trajectories of the two models from the same initial point and their final positions relative to the local minimum, we can effectively compare them. SEMA ultimately converges rapidly to the position closest to the local minimum, while EMA remains trapped in a flat local basin. This effectively demonstrates that SEMA can converge faster and more efficiently.

**Decision Boundary.** We trained a 2-layer MLP classifier using the SGD optimizer with a fixed learning rate of 0.01 on a binary classification dataset from sklearn (Pedregosa et al., 2011). The dataset consisted of circular and moon-shaped data points, with 50 labeled samples represented by red or yellow triangles and the remaining samples as grey circles for testing purposes. We compared the performance of the baseline model, EMA, and SEMA. When plotting the decision boundaries (He et al., 2018), we

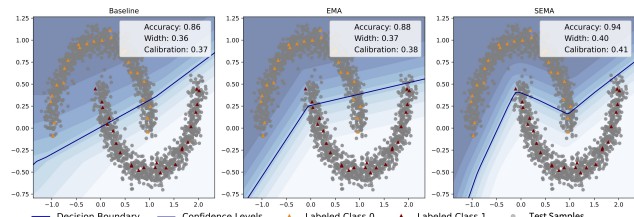

Figure A2: Illustration of the baseline, EMA, and SEMA on Two Moons Dataset with 50 labeled samples (triangle red/yellow points) with others as testing samples (grey points) in training a 2-layer MLP classifier. We evaluate the performance by computing top-1 accuracy, decision boundary width, and prediction calibration.

used smooth, solid lines to represent the boundaries ( 4,  A2). Each algorithm was distinguished by using distinct colors. The transition of the test samples from blue to grey represented the confidence of the network predictions. Furthermore, SEMA can be further compared and evaluated according to the evaluation indicators. One is accuracy, that is, the ability to divide test samples; the other is Calibration, the closer the decision boundary, the less trustworthy it will be, and the lower the classification accuracy will be; the third is the width of the decision boundary, the wider the decision boundary, the more robust it will be. According to the intuitive comparison, SEMA can achieve the most accurate classification and is superior to EMA and the baseline model in all evaluation indicators, effectively indicating that SEMA has higher performance and robustness.

# D EXTENSIVE RELATED WORK

## D.1 OPTIMIZERS

With the advent of BP (Rumelhart et al., 1986) and SGD (Sinha & Griscik, 1971) with mini-batch training (Bishop, 2006), optimizers have become an indispensable component in the training process of DNNs. Various mainstream optimizers leverage momentum techniques (Sutskever et al., 2013) to accumulate gradient statistics, which are crucial for improving the convergence and performance of DNNs. In addition, adaptive learning rates such as those found in Adam variants (Kingma & Ba, 2014; Liu et al., 2020) and acceleration schemes (Kobayashi, 2020) have been used to enhance these capabilities further. SAM method (Foret et al., 2021) works by searching for a flatter region where the training losses in the estimated neighborhood are minimized through min-max optimizations. Its variants aim to improve training efficiency from various perspectives, such as gradient decomposition (Zhuang et al., 2022) and training costs (Du et al., 2021; 2022) (Liu et al., 2022a). To expedite training, large-batch optimizers like LARS (Ginsburg et al., 2018) for SGD and LAMB (You et al., 2020) for AdamW (Loshchilov & Hutter, 2019) have been proposed. These optimizers adaptively adjust the learning rate based on the gradient norm to facilitate faster training. Adan (Xie et al., 2023) introduces Nesterov descending to AdamW, bringing improvements across popular computer vision and natural language processing applications. Moreover, a new line of research has proposed plug-and-play optimizers such as Lookahead (Zhang et al., 2019; Zhou et al., 2021) and Ranger (Wright, 2019). These optimizers can be combined with existing inner-loop optimizers (Zhou et al., 2021), acting as the outer-loop optimization to improve generalization and performance while allowing for faster convergence.

## D.2 WEIGHT AVERAGING

In stark contrast to gradient momentum updates in optimizers, weight averaging (WA) techniques such as SWA (Izmailov et al., 2018) and EMA (Polyak & Juditsky, 1992) are commonly employed during DNN training to enhance model performance further. Test-time WA strategies, including SWA variants (Maddox et al., 2019) and FGE variants (Guo et al., 2023; Garipov et al., 2018), employ the heuristic of ensembling different models from multiple iterations (Granziol et al., 2021) to achieve flat local minima and thereby improve generalization capabilities. Notably, TWA (Li et al., 2023c) has improved upon SWA by implementing a trainable ensemble. Another important weighted averaging technique targeted explicitly at large models is Model Soup (Wortsman et al., 2022), which leverages solutions obtained from different fine-tuning configurations. Greedy Soup improves model performance by sequentially greedily adding weights. When applied during the training phase, the EMA update can significantly improve the performance and stability of existing optimizers across various domains. EMA is an integral part of certain learning paradigms, including popular semi-supervised learning methods such as FixMatch variants (Sohn et al., 2020), and self-supervised learning (SSL) methods like MoCo variants (He et al., 2020; Chen et al., 2021), and BYOL variants (Grill et al., 2020). These methods utilize a self-teaching framework, where the teacher model parameters are the EMA version of student model parameters. In the field of reinforcement learning, A3C (Mnih et al., 2016) employs EMA to update policy parameters, thereby stabilizing the training process. Furthermore, in generative models like diffusion (Karras et al., 2023), EMA significantly contributes to the stability and output distribution. Recent efforts such as LAWA (Kaddour, 2022) and PSWA (Guo et al., 2022) have explored the application of EMA or SWA directly during the training process. However, they found that while using WA during training can accelerate convergence, it does not necessarily guarantee final performance gains. SASAM (Kaddour et al., 2022) combines the complementary merits of SWA and SAM to achieve local flatness better. Despite these developments, the universal applicability and ease of migration of these WA techniques make them a crucial focus for ongoing innovation. This paper aims to improve upon EMA by harnessing the historical exploration of a single configuration and prioritizing training efficiency to achieve faster convergence.

## D.3 REGULARIZATIONS

In addition to the optimizers, various regularization techniques have been proposed to enhance the generalization and performance of DNNs. Techniques such as weight decay (Andriushchenko et al., 2023) and dropout variants (Srivastava et al., 2014; Huang et al., 2016) are designed to control the

complexity of the network parameters and prevent overfitting, which have been effective in improving model generalization. These techniques, including EMA, fall under network parameter regularizations. Specifically, the EMA has shown to effectively regularize Transformer (Devlin et al., 2018; Touvron et al., 2021) training across both computer vision and natural language processing scenarios (Liu et al., 2022b; Wightman et al., 2021). Another set of regularization techniques aims to improve generalizations by modifying the data distributions. These include label regularizers (Szegedy et al., 2016) and data augmentations (DeVries & Taylor, 2017). Data-dependent augmentations like Mixup variants (Zhang et al., 2018; Yun et al., 2019; Liu et al., 2022d) and data-independent methods such as RandAugment variants (Cubuk et al., 2019; 2020) increase data capacities and diversities. These methods have achieved significant performance gains while introducing negligible additional computational overhead. Most regularization methods, including our proposed SEMA, provide 'free lunch' solutions. They can effectively improve performance as a pluggable module without incurring extra costs. In particular, SEMA enhances generalization abilities as a plug-and-play step for various application scenarios.

