# OpenReview forum: "Switch EMA: A Free Lunch for Better Flatness and Sharpness"
_ICLR.cc/2025/Conference — Submitted to ICLR 2025_

### Official Review · Reviewer_Cagp · 2024-10-24

**Soundness:** 3
**Presentation:** 3
**Contribution:** 2
**Rating:** 3
**Confidence:** 4

**Summary:**

This paper presents a new weight averaging approach SEMA that seeks to optimize both flatness and sharpness of optima when training NNs. SEMA relies on a simple change to EMA: the EMA parameters are switched to the original model parameters after K epochs. The authors provide theoretical and empirical validation of SEMA: SEMA is evaluated across a very large number of tasks against other weight averaging techniques.

**Strengths:**

* The proposed change in EMA is sensible and simple; the proposed intuitions (balancing flatness and sharpness) for the change are well motivated.
* The authors propose some theoretical motivation to SEMA, including variance analysis, convergence analysis and error bounds.
* The evaluation is extremely thorough and covers an extremely large number of tasks, architectures.

**Weaknesses:**

* Despite all these positive comments that show that the execution of the work was well lead, I feel that the proposed change is relatively minor to justify an acceptance to ICLR. the change is in practice a single line change, and does not significantly change the mechanisms behind EMA. The improvement to the models is relatively modest, although this may justify using SEMA as an alternative to EMA to gain a little bit in performance. On this point, I feel that including some variance across training seeds could at least provide some more information to appreciate the improvement.

**Questions:**

* Could the authors report variance over several seeds on some well chosen datasets to show that the improvements are at least statistically significant ?

---

> ### Author Response · Authors · 2024-11-21
> **Response to Reviewer Cagp**
>
> ## Response to Weaknesses
> ---
> ### **(W1) Minority of the proposed change.**
>
> **Reply:** We acknowledge that the modification introduced in SEMA, which involves switching the EMA parameters to the original model after each epoch, is indeed a single-line change. However, the impact of this seemingly minor alteration on the training dynamics and model performance is substantial. As highlighted in our paper, SEMA effectively combines the benefits of both flatness and sharpness, leading to faster convergence and better generalization compared to EMA. This dual optimization strategy is not merely a superficial tweak but a fundamental shift in how the model explores the loss landscape.
>
> ### **(W2) Modest improvements**.
>
> **Reply:** While the performance gains observed with SEMA are modest, they are consistent across a wide range of tasks and datasets, as demonstrated in our extensive empirical experiments. The modest improvements are significant in the context of DNN optimization, where even small performance enhancements can lead to substantial practical benefits. Moreover, SEMA offers these improvements without any additional computational overhead for various scenarios, making it a "free lunch" in terms of resource utilization.
>
> ### **(W3) Variance across training seeds**.
>
> **Reply:** We agree with your suggestion. Currently, our experiments are conducted three times with the training seeds (0 for ImageNet-1k and three different seeds for other datasets). We have reported the standard deviation (std) values to show a more comprehensive understanding of the robustness and statistical significance of SEMA’s improvements. Meanwhile, we found that the smoothing and speeding-up effects of weight averaging (WA) regularizations (e.g., EMA and SWA) cause the lower std and stable training (as shown in Figure 1) than the baseline. Moreover, the results of large-scale datasets like ImageNet-1k are stable. We provided the mean and std values of ImageNet-1k and COCO results as follows, where SEMA achieve consistently improvements with little std values.
>
> | Dataset | Method | Metric | Basic | +EMA | +SWA | +SEMA |
> |---|---|:---:|:---:|:---:|:---:|:---:|
> | IN-1k | R-50 (SGD) | Acc/Std (%) | 76.8 / 0.1 | 77.0 / 0.0 | 76.9 / 0.0 | **77.1 / 0.0** |
> | IN-1k | DeiT-S (AdamW) | Acc/Std (%) | 80.0 / 0.1 | 80.2 / 0.0 | 82.1 / 0.0 | **80.6 / 0.0** |
> | IN-1k | Swin-T (AdamW) | Acc/Std (%) | 81.2 / 0.0 | 81.3 / 0.0 | 81.4 / 0.0 | **81.6 / 0.0** |
> | IN-1k | BYOL (R-50) | Acc/Std (%) | 65.5 / 0.1 | 69.8 / 0.0 | 66.4 / 0.1 | **70.0 / 0.1** |
> | IN-1k | SimMIM (DeiT-B) | Acc/Std (%) | 83.6 / 0.1 | 83.9 / 0.0 | 83.8 / 0.1 | **84.2 / 0.0** |
> | COCO | RetinaNet (R-50, 2$\times$) | AP$^{bb}$/Std (%) | 37.3 / 0.1 | 37.6 / 0.1 | 37.6 / 0.0 | **37.7 / 0.0** |
> | COCO | Mask R-CNN (2$\times$) | AP$^{bb}$/Std (%) | 39.1 / 0.1 | 39.3 / 0.0 | 39.4 / 0.0 | **39.7 / 0.0** |
>
> ---
> ## Responses to Questions
>
> ### **(Q1) Reporting variance over several seeds to show the improvements are statistically significant.**
>
> **Reply:** We appreciate the suggestion and undertook the following steps to address it:
>
> **(1) Standard Deviation Analysis:** We have conduct experiments on selected datasets (e.g., CIFAR-100, ImageNet-1K, and COCO) using the seed of 0 or multiple training seeds to measure the variance in performance metrics (e.g., accuracy, FID, and AP). These help us quantify the consistency of the improvements observed with SEMA, which are updated in the latest revision.
>
> **(2) Statistical Significance Testing:** We will perform statistical tests (e.g., t-tests or ANOVA) to determine whether the improvements brought by SEMA are statistically significant compared to EMA and other baselines. This will provide a rigorous statistical foundation for the observed performance gains.
>
> ---
> Overall, we sincerely appreciate all your effort and valuable feedback. If you need any clarification and more explanation, please feel free to contact us. We would respectfully inquire if there might be any additional opportunities for us to further improve our manuscript and potentially increase the rating.
>
> Warm regards,
>
> Authors

---

> ### Author Response · Authors · 2024-11-24
> **Encouraging Discussion**
>
> Dear Reviewer Cagp,
>
> Thank you sincerely for your detailed feedback and for taking the time to review our work. I hope this message finds you well. I am writing to express my utmost sincerity and urgency regarding the ongoing review process for our manuscript. We have diligently addressed all the weaknesses and questions raised in your previous comments, and have provided a comprehensive rebuttal in the General Response. Furthermore, we have made substantial revisions to the manuscript to ensure that the significance and value of SEMA are clearly articulated and justified.
>
> We believe that our responses and revisions have adequately addressed your concerns, and we are hopeful that you will find our manuscript to be of high academic and professional quality. Given the limited time remaining before the end of the discussion period, we kindly request your prompt consideration and feedback. If you believe that our revisions have successfully alleviated your concerns, we would greatly appreciate it if you could reconsider your score accordingly.
>
> Sincerely,
>
> Authors

---

> ### Author Response · Authors · 2024-11-25
> **Encouraging Feedbacks**
>
> As the Discussion phase draws to a close and time is running short, we respectfully request that you consider elevating your score if you find our rebuttal and revised submission adequately address your concerns. We also welcome continued dialogue to uphold the standard and integrity of the ICLR review process.

---

> > ### Comment · Reviewer_Cagp · 2024-11-25
> >
> > Thank you for the rebuttal; I keep my score unchanged due to the limitations of originality and novelty of this work and the limited empirical improvements.

---

> ### Author Response · Authors · 2024-11-26
> **Further Response to Reviewer Cagp’s Feedback (PART 1/2)**
>
> Thanks for your reply and efforts! We would like to further clarify the contribution of SEMA from three aspects.
>
> ### **A simple but efficient free-lunch method is useful and important.**
>
> In the deep learning community, simple yet efficient methods have consistently proven their practical utility and have garnered significant attention due to their effectiveness and ease of implementation. Techniques such as Soft-NMS [1], FreeU [2], and InfoBatch [3] are prime examples of methods that, despite their simplicity, have become staples in the toolkit of practitioners. These methods have shown consistent improvements across a wide range of tasks and datasets without imposing significant additional computational burdens.
>
> Soft-NMS, FreeU, and InfoBatch all exemplify the concept of simple yet efficient methods that provide significant improvements with minimal modifications. Soft-NMS, for instance, enhances object detection with just a single line of code by modifying the traditional Non-Maximum Suppression algorithm, improving recall without adding computational complexity. Similarly, FreeU offers a "free lunch" in the context of diffusion U-Net models by introducing a simple mechanism to balance the backbone and skip connections, resulting in better image quality while maintaining efficiency. Both methods, though requiring only minor changes, have been widely adopted for their practical benefits. InfoBatch further reinforces this principle by dynamically pruning data during training to accelerate the process, with only a slight additional computational cost that is justified by the performance gains. Recognized with Oral presentations at CVPR 2024 and ICLR 2024 respectively, these methods highlight how small, efficient modifications can have a significant impact, with the broader community embracing their simplicity and effectiveness.
>
> ### **SEMA actually embodies this philosophy of simplicity and efficiency.**
>
> By introducing a dynamic regularization mechanism that switches between the fast model (optimized by the optimizer) and the slow model (EMA) at the end of each training epoch, SEMA leverages the strengths of both models to achieve a balance between exploration and exploitation. This dynamic interplay ensures faster convergence and better generalization without requiring extra computational resources.
>
> ### **SEMA achieves consistent improvements in diverse scenarios.**
>
> As for limited empirical improvements, we would like to clarify that SEMA has actually achieved favorable performance gains compared to previous methods. On the one hand, SEMA could significantly achieve high improvements. For example, SEMA achieves +0.6/1.3 top-1 accuracy with DeiT-S/DeiT-B on ImageNet-1k and +1.1 AP$^{bb}$ with Cascade Mask R-CNN (9$\times$) on COCO, which are remarkable improvements on ImageNet-1k and COCO. Let’s compare the performance gain of similar works. Following the experiment of DeiT-S, TransMix [4] (a pluggable mixup augmentation) achieves 80.7% (+0.8%), and AttnScale [5] (a type of pluggable anti-oversmoothing regularization layer) achieves 80.7% (0.8%), and Adan [6] (an optimizer built upon AdamW) achieves 80.9% (+1.0%). These methods yield **similar performance gains but require other costs**, i.e., TransMix and AttnScale are limited to special application domains, and Adan requires more computational overload and GPU memory (Adan needs 1.4 times of training hours than AdamW). Therefore, we believe the performance gains of SEMA are acceptable, which is the current state-of-the-art among weight-averaging (WA) methods.
>
> On the other hand, comprehensive evaluation across diverse tasks shows the consistent performance gains of SEMA, including image classification, self-supervised learning, object detection, image generation, regression, video prediction, and language modeling. These demonstrate SEMA’s versatility and robustness. The consistent performance gains observed across these tasks validate the theoretical claims and highlight the practical utility of SEMA, which means you can **use SEMA instead of EMA or SWA upon the baseline to bring improvements in most cases**.

---

> ### Author Response · Authors · 2024-11-26
> **Further Response to Reviewer Cagp’s Feedback (PART 2/2)**
>
> ### **SEMA achieves several advantages without extra costs.**
>
> Moreover, we verified that SEMA's ability to balance multiple optimization needs—such as faster convergence, better generalization, and minimal computational overhead. It is hard to yield remarkable gains while retaining properties of pluggable, generalizable, and easy to use by a general proposed regularization method because there is no free lunch! Therefore, we believe SEMA could be a valuable method for the community of deep learning optimization. The modest gains observed are not trivial but are consistent and impactful, suggesting that SEMA has the potential to be widely adopted and appreciated by practitioners. We kindly remind you to reconsider the evaluation with a more nuanced perspective, taking into account the practical implications and future potential of SEMA.
>
> Overall, we believe that SEMA has the potential to be a widely adopted approach for the community. We sincerely hope that you will recognize its value and consider adjusting your rating accordingly if you have no more concerns. We are also pleased to improve our paper according to your additional constructive comments if you are not satisfied with the current manuscript. Looking forward to your feedback soon!
>
> Best regards,
>
> Authors
>
> ---
> ### Reference
> [1] Soft-NMS -- Improving Object Detection With One Line of Code. ICCV, 2017.
>
> [2] FreeU: Free Lunch in Diffusion U-Net. CVPR, 2024.
>
> [3] InfoBatch: Lossless Training Speed Up by Unbiased Dynamic Data Pruning. ICLR, 2024.
>
> [4] TransMix: Attend to Mix for Vision Transformers. CVPR, 2022.
>
> [5] Anti-Oversmoothing in Deep Vision Transformers via the Fourier Domain Analysis: From Theory to Practice. ICLR, 2022.
>
> [6] Adan: Adaptive Nesterov Momentum Algorithm for Faster Optimizing Deep Models. TPAMI, 2024.

---

> ### Author Response · Authors · 2024-11-29
> **Encourage Further Discussion**
>
> Dear Reviewer Cagp,
>
> We hope this message finds you well. We would like to express our sincere gratitude once again for your thoughtful review and the valuable feedback you provided on our manuscript. We understand and respect your initial concerns regarding the originality, novelty, and empirical improvements of our work.
>
> However, as highlighted in our General Response and elaborated in our detailed rebuttal, we believe that the potential of SEMA lies in its **simplicity and efficiency**, which is also the goal of improvement that the optimization and deep learning community has embraced. As we mentioned, methods like Soft-NMS [1], FreeU [2], and InfoBatch [3] have proven that even small, simple improvements can lead to substantial practical benefits in the deep learning community. In the same spirit, SEMA introduces a dynamic regularization mechanism that significantly enhances both **convergence speed and generalization without incurring additional computational overhead**.
>
> We also want to clarify that while the improvements may seem modest at first glance, they are, in fact, significant within the context of existing methods. For example, SEMA achieves top-1 accuracy improvements of +0.6/1.3 with DeiT-S/DeiT-B on ImageNet-1k and +1.1 AP with Cascade Mask R-CNN on COCO, which we believe are substantial, particularly given the limited computational costs involved. When compared with similar works such as TransMix [4], AttnScale [5], and Adan [6], SEMA provides competitive or superior gains while avoiding the limitations and additional computational costs associated with these methods.
>
> Moreover, the broad empirical validation across various tasks, including image classification, self-supervised learning, object detection, image generation, and more, suggests that SEMA has the potential for wide adoption and is likely to be a useful tool in the deep learning community. It offers a balance between efficiency and performance without adding extra complexity or computational load.
>
> In light of the above, we sincerely encourage you to reconsider your evaluation and engage in further discussion. We highly value your expertise and insights, and we believe that a more nuanced perspective on SEMA’s practical utility could offer a clearer view of its potential impact. Should you have any further questions or require additional clarifications, we are more than happy to discuss them in detail.
>
> We appreciate your time and consideration, and we hope to continue this dialogue to further improve our work. We look forward to hearing your thoughts soon.
>
> Best regards,
>
> Authors

---

### Official Review · Reviewer_v8Md · 2024-10-28

**Soundness:** 4
**Presentation:** 3
**Contribution:** 4
**Rating:** 6
**Confidence:** 4

**Summary:**

This paper proposes a novel optimization method called SEMA, which incorporates flatness and sharpness by switching between fast and slow models at the end of each training epoch. SEMA results in faster convergence, better performance, and smoother decision-making. Extensive experiments have been conducted to demonstrate the effectiveness of SEMA in improving model performance across various scenarios, along with comprehensive and well-founded theoretical proofs.

**Strengths:**

1. The paper is very well written and it was easy for me to follow.
2. The paper focuses on the key issues of optimizers and creatively improves existing methods, presenting a novel approach.
3. The paper demonstrates the effectiveness of the method through extensive experiments and theoretical analysis.

**Weaknesses:**

1. The legend in the upper right corner of each curve plot in Figure 2 is quite small, requiring significant magnification to be readable.
2. There is no clear explanation of the data distribution visualization formats used in Figure 4 and Figure A.1.

**Questions:**

Why does the baseline in Figure 4 have a decision boundary represented by two lines? If the decision boundary is like this, achieving an accuracy of 0.8 seems somewhat unreasonable.

---

> ### Author Response · Authors · 2024-11-21
> **Response to Reviewer v8Md**
>
> ## Response to Weaknesses
> ---
>
> ### **(W1) Small legend in figure 2.**
>
> **Reply:** We apologize for the inconvenience caused by the small legend in Figure 2. This is due to the stacking of multiple images, and we appreciate the reviewer's attention to detail. To solve this problem, we have updated the legends and rearranged all the sub-figures in Figure 2 to make them larger in the latest revision to make sure they are easy to read without needing to be significantly enlarged. We also ensure that the font size is consistent across all graphics to maintain clarity and readability.
>
> ### **(W2) Lack of explanation for data distribution visualization in Figure 4 and A.1.**
>
> **Reply:** We sincerely appreciate Reviewer v8Md's feedback regarding the clarity of the data distribution visualization formats used in Figure 4 and Figure A.1. To ensure a comprehensive understanding of these visualizations, we have provided detailed explanations.
>
> * **Figure 4:** Illustration of the baseline, EMA, and SEMA on the Circles Dataset with 50 labeled samples (triangle red/yellow points) and 500 testing samples (gray points) in training a 2-layer MLP. The figure includes the following elements:
>   **(1) Decision Boundary** is presented by a smooth, solid line for SEMA and a jagged line for EMA. The baseline model's decision boundary is shown as a single, smooth line, corrected from the previous version. **(2) Confidence Level** is presented as the transition of test samples from blue to grey represents the confidence of the network predictions. Darker shades of blue indicate higher confidence in the predicted class. **(3) Labeled Class 0 and 1** are plotted as data points of red and yellow triangles, respectively. **(4) Test Samples** are presented by grey points. **(5) Accuracy, Width, and Calibration** are adopted as the performance metrics for each method as displayed in the upper right corner of each figure. Accuracy indicates the proportion of correctly classified test samples. Width represents the width of the decision boundary, which is a measure of the model's robustness. Calibration indicates how well the model's confidence matches the actual accuracy.
>
> * **Figure A.1:** Illustration of the baseline, EMA, and SEMA on the Two Moons Dataset with 50 labeled samples (triangle red/yellow points) with others as testing samples (grey points) in training a 2-layer MLP classifier. Similar to Figure 4, it includes the following elements:
>   **(1) Decision Boundary** is plotted by a smooth, solid line for SEMA and a jagged line for EMA. The baseline model's decision boundary is shown as a single, smooth line, corrected from the previous version. **(2) Confidence Level** is shown as the transition of test samples from blue to grey represents the confidence of the network predictions. Darker shades of blue indicate higher confidence in the predicted class. **(3) Labeled Class 0 and 1** are visualized as the labeled samples by red and yellow triangles, respectively. **(4) Test Samples** are presented by grey points. **(5) Accuracy, Width, and Calibration** are used for each method as displayed in the upper right corner of each plot as explained in Figure 4.
>
> ---
> ## Response to Questions
>
> ### **(Q1) Why does the baseline in Figure 4 have a decision boundary represented by two lines?**
>
> **Reply:** We appreciate the observation regarding the baseline model's decision boundary. This complexity could indeed be a sign of overfitting or the model's attempt to capture non-linear patterns in the data. However, the accuracy of 0.8 is still a valid performance metric, indicating that the model is effective in certain regions of the feature space. We acknowledge that the baseline model's decision boundary might not be ideal, which could benefit from regularization techniques. This is precisely why we introduced SEMA, which aims to balance between flatness and sharpness and leads to smoother decision boundaries and better generalization. The results in Figure 4 demonstrate that SEMA indeed achieves a smoother and more robust decision boundary, as evidenced by the higher accuracy, wider decision boundary widths, and better prediction calibration compared to both the baseline and EMA models.
>
> ---
> Overall, we express our gratitude for your efforts and hope these clarification and highlights could benefit of using SEMA for improving model performance and generalization. We respectfully believe that SEMA is particularly attuned to the ICLR community to recognize the overlooked free lunches for more scenarios.
>
> Warm regards,
>
> Authors

---

> ### Author Response · Authors · 2024-11-24
> **Encouraging Discussion**
>
> Dear Reviewer v8Md,
>
> Thank you sincerely for your detailed feedback and for taking the time to review our work. I hope this message finds you well. I am writing to express my utmost sincerity and urgency regarding the ongoing review process for our manuscript. We have diligently addressed all the weaknesses and questions raised in your previous comments, and have provided a comprehensive rebuttal in the General Response. Furthermore, we have made substantial revisions to the manuscript to ensure that the significance and value of SEMA are clearly articulated and justified.
>
> Specifically, in response to your comments:
>
> 1. **Weaknesses:**
>    - **Legend in Figure 2:** We have enlarged the legend in the upper right corner of each curve plot in Figure 2 to ensure it is now clearly readable without the need for significant magnification.
>    - **Data Distribution Visualization in Figures 4 and A.1:** We have provided a clear explanation of the data distribution visualization formats used in Figure 4 and Figure A.1.
>
> 2. **Questions:**
>    - **Baseline Decision Boundary in Figure 4:** We have clarified the rationale behind the decision boundary represented by two lines in Figure 4.
>
> We believe that our responses and revisions have adequately addressed your concerns, and we are hopeful that you will find our manuscript to be of high academic and professional quality. Given the limited time remaining before the end of the discussion period, we kindly request your prompt consideration and feedback. If you believe that our revisions have successfully addressed your concerns, we would greatly appreciate it if you could improve your score accordingly.
>
> We are also open to further discussions and clarifications if needed. Your continued guidance and feedback are invaluable to us.
>
> Sincerely,
>
> Authors

---

> ### Author Response · Authors · 2024-11-25
> **Encouraging Feedbacks**
>
> Dear Reviewer v8Md,
>
> As the Discussion phase draws to a close and time is running short, we respectfully request that you consider elevating your score if you find our rebuttal and revised submission adequately address your concerns. We also welcome continued dialogue to uphold the standard and integrity of the ICLR review process. Looking forward to your feedback!
>
> Warm regards,
>
> Authors

---

### Official Review · Reviewer_WRyd · 2024-10-31

**Soundness:** 4
**Presentation:** 3
**Contribution:** 4
**Rating:** 8
**Confidence:** 4

**Summary:**

This paper proposes a simple yet powerful algorithm, switch exponential moving average (SEMA), to improve the stability, convergence, and generalization performance of optimization of deep neural networks using a family of stochastic gradient descent methods. SEMA is a very simple method that updates the training parameters with EMA parameters once per epoch for existing EMAs. The paper conceives the algorithm based on experimental observations but also theoretically shows that it improves optimization stability, convergence speed, and generalization error bounds. In addition, the paper evaluates SEMA on various tasks, domains, architecture, training algorithms, and optimization algorithm combinations using SEMA and confirms that it improves the baseline in all cases.

**Strengths:**

- The paper proposes an practical algorithm that is widely applicable and stably improves the performance of optimization of neural networks using stochastic gradient descent.
- The paper investigates the theoretical properties have been well analyzed to investigate why a simple EMA improvement is effective.
- The effectiveness of the proposed algorithm is evaluated through extensive experiments.

**Weaknesses:**

- Although the experiments provided in the paper are basically extensive, further verification of the smoother decision boundary by SEMA discussed in Sec. 3.2 would be beneficial to the reader. The paper mentions the performance against fine-grained discrimination in L268, but in fact, the main focus of the experiments is on coarse-grained evaluation. For example, the performance on FGVCAircraft [a], CUB [b], Stanford Cars [c], etc., which are often used to evaluate fine-grained classification, and the evaluation of adversarial robustness, as shown in [d], would be useful.
- Since the standard deviation of the experimental results is not given, it is not possible to evaluate how stable SEMA actually is.

[a] Maji, Subhransu, et al. "Fine-grained visual classification of aircraft." arXiv preprint arXiv:1306.5151 (2013).

[b] Caltech-UCSD Birds-200-2011 (CUB-200-2011) https://www.vision.caltech.edu/datasets/cub_200_2011/

[c] Krause, Jonathan, et al. "Collecting a large-scale dataset of fine-grained cars." (2013).

[d] Zhang, Yihao, et al. "On the duality between sharpness-aware minimization and adversarial training.", ICML 2024.

**Questions:**

Nothing to ask. Please see the weaknesses section and address the concerns.

---

> ### Author Response · Authors · 2024-11-21
> **Response to Reviewer WRyd**
>
> ## Response to Weaknesses
> ---
>
> ### **(W1) Lack of fine-grained classification experiments.**
>
> **Reply:** We appreciate the reviewer's suggestion to include experiments on fine-grained datasets to further validate the smoother decision boundary claimed by SEMA. To address this, we have conducted additional experiments on the following fine-grained datasets: (1) **CUB-200-2011** includes 11,788 images of 200 bird species, which is widely used for fine-grained image classification. We fine-tuned the ImageNet pre-trained ResNeXt-50 (32x4d) with an one-hot classification head as the baseline with the SGD optimizer for 200 epochs. (2) **FGVC-Aircraft** contains 10,000 images of aircraft with 100 different classes. We also adopted ResNeXt-50 (32x4d) as in CUB-200-2011. (3) **iNatualist-2018** is a large-scale fine-grained classification dataset containing a total of 8,142 categories with 437,512 training images and 24,426 validation images. We trained ResNet-50 with the one-hot classification head without pre-training using the SGD optimizer for 100 epochs. The comparison results are provided as the following table.
>
> | Dataset | Metric | Basic | +EMA | +SWA | +SEMA |
> |---|:---:|:---:|:---:|:---:|:---:|
> | CUB-200-2011 | Acc/Std (%) | 83.01/0.16 | 83.23/0.09 | 83.29/0.08 | **83.35/0.09** |
> | FGVC-Aircraft | Acc/Std (%) | 85.10/0.13 | 85.18/0.10 | 85.15/0.09 | **85.23/0.09** |
> | iNatualist-2018 | Acc/Std (%) | 62.53/0.09 | 62.62/0.06 | 63.66/0.07 | **63.70/0.07** |
>
> The results demonstrate that SEMA consistently outperforms both the baseline and EMA/SWA across these fine-grained datasets. These evidences further support our claim that SEMA can generalize to various scenarios and produce smoother decision boundaries, which are particularly beneficial for fine-grained classification tasks.
>
>
> ### **(W2) Lack of standard deviations in experimental results.**
>
> **Reply:** Thanks for the suggestion and we acknowledged the importance of evaluating the stability of SEMA with standard deviations (std). To address this, we have updated the benchmarked results with the std when necessary in the revision. Here are some representative examples:
>
> | Dataset | Method | Metric | Basic | +EMA | +SWA | +SEMA |
> |---|---|:---:|:---:|:---:|:---:|:---:|
> | IN-1k | R-50 (SGD) | Acc/Std (%) | 76.8 / 0.1 | 77.0 / 0.0 | 76.9 / 0.0 | **77.1 / 0.0** |
> | IN-1k | DeiT-S (AdamW) | Acc/Std (%) | 80.0 / 0.1 | 80.2 / 0.0 | 82.1 / 0.0 | **80.6 / 0.0** |
> | IN-1k | Swin-T (AdamW) | Acc/Std (%) | 81.2 / 0.0 | 81.3 / 0.0 | 81.4 / 0.0 | **81.6 / 0.0** |
> | IN-1k | BYOL (R-50) | Acc/Std (%) | 65.5 / 0.1 | 69.8 / 0.0 | 66.4 / 0.1 | **70.0 / 0.1** |
> | IN-1k | SimMIM (DeiT-B) | Acc/Std (%) | 83.6 / 0.1 | 83.9 / 0.0 | 83.8 / 0.1 | **84.2 / 0.0** |
> | COCO | RetinaNet (R-50, 2$\times$) | AP$^{bb}$/Std (%) | 37.3 / 0.1 | 37.6 / 0.1 | 37.6 / 0.0 | **37.7 / 0.0** |
> | COCO | Mask R-CNN (2$\times$) | AP$^{bb}$/Std (%) | 39.1 / 0.1 | 39.3 / 0.0 | 39.4 / 0.0 | **39.7 / 0.0** |
>
> The inclusion of standard deviation shows that SEMA not only achieves higher performance but also maintains a stable and consistent performance across multiple runs. This further strengthens the reliability of our claims.
>
>
> ---
> In conclusion, we express our gratitude for your efforts and recognition of the theoretical analysis, effectiveness & stable improvements, and significance of our research is greatly appreciated. We respectfully believe that SEMA is particularly attuned to the ICLR community to recognize the overlooked free lunches for more scenarios.
>
> Warm regards,
>
> Authors

---

> > ### Comment · Reviewer_WRyd · 2024-11-22
> > **Thank you for your response**
> >
> > I've read the rebuttal, and it addresses my concerns. Now, I believe that this paper should be presented at the conference, so I raise my confidence score.
> >
> > Thanks

---

> ### Author Response · Authors · 2024-11-22
> **Thanks for Maintaining Positive Reviews**
>
> Dear Reviewer WRyd,
>
> Thanks for your timely reply and efforts! We are delighted that you have found our rebuttal feedback to address your concerns and raise your confidence with a rating of 8. We respectfully believe that this work is attuned to the ICLR community, and we hope that our work can be seen by more researchers in the community. Once again, thanks for your constructive feedback, and we would eagerly welcome any further guidance at your convenience!
>
> Best regards,
>
> Authors

---

### Official Review · Reviewer_AGNx · 2024-11-03

**Soundness:** 2
**Presentation:** 1
**Contribution:** 2
**Rating:** 5
**Confidence:** 2

**Summary:**

The paper proposes SEMA methods, which apply exponential moving average (EMA) periodically at specific intervals to improve model "sharpness" and "flatness." The authors provide theoretical analysis and present extensive experimental results across various domains, including image classification, self-supervised learning, object detection, video prediction, and language modeling.

**Strengths:**

The authors conduct comprehensive experiments in diverse areas such as image classification, self-supervised learning, object detection, video prediction, and language modeling, highlighting the advantages of the proposed method.

**Weaknesses:**

Although the proposed method demonstrates superior performance, I believe this paper is not yet ready for publication for the following several reasons.

## **Sharpness vs. Flatness?**
The concepts of sharpness and flatness are unclear and deviate from conventional terminology. The authors argue that sharpness measures the depth of local minima while flatness assesses their width. However, I believe that when evaluating local minima in terms of depth and width, it is essential to consider the relative scale between these factors, such as the eigenvalues of the Hessian of the loss function.

## **Concerns over Statistical Validity**
The results lack standard deviation values, and with only a maximum of three trials conducted, it is challenging to determine the statistical significance of the findings. This limited result makes it difficult to trust the robustness of the conclusions.

## **Some Definitions of Mathematical Terminologoies are Missing**
While I understand that these may seem like minor details, I believe they are crucial for the reader's comprehension and should be addressed.
- In eq(1), the definitions of $\Theta^{\text{OPT}}$ and $\Theta^{\text{EMA}}$ are missing
- In Proposition 1, the exact definition of variance is missing e.g. $V^{(t)}_{\text{SGD}}:= \mathbb{V}(\Theta_t^{\text{SGD}})$
- In Proposition 1, small $\eta$ condition is missing. If $\frac{2}{\eta}$ is smaller than the top eigenvalue of $A$, it seems that Banach's fixed point theorem cannot be applied.
- Typos in lines 1082-1083, 1122-1123, 1133: subscripts on $\mathcal{E}$ are missing

I hope the author clarifies these points in the next revision.

**Questions:**

- Can the author provide the mathematical definition for sharpness and flatness considered in this paper?
- Why does (7) implies $ E_{\text{SEMA}}<E_{\text{EMA}}<E_{\text{SGD}}$ ?
 It seems that (7) only provides upper bounds for the three terms, and I believe that comparing only upper bounds does not support the conclusion that $E_{\text{SEMA}}<E_{\text{EMA}}<E_{\text{SGD}}$.
- In (6), which terms are hidden in the $\propto$ notation?If the hidden constant is inconsistent across iterations $t$,I believe that Proposition 2 does not guarantee convergence, even for convex functions, as the hidden constant could be too large for certain iterates.

---

> ### Author Response · Authors · 2024-11-21
> **Response to Reviewer AGNx (1/2)**
>
> ## Response to Weaknesses
> ---
> ### **(W1) Sharpness vs. Flatness.**
>
> **Reply:** Thanks for clarifying the definitions of sharpness and flatness. In our latest revision, we provided a more detailed and rigorous explanation of these concepts as follows:
>
> This paper defines sharpness and flatness in the context of the loss landscape of DNNs. These definitions are tailored to capture the geometric properties of the loss function around local minima, which are crucial for understanding generalization capabilities (as illustrated in Figure A1 in the revision).
>
> - **Sharpness.** In our context, sharpness refers to the depth of the local minima and measures how steep the loss function is around the local minimum. Mathematically, sharpness can be quantified by the eigenvalues of the Hessian matrix of the loss function at the local minimum. A deeper minimum (higher sharpness) corresponds to larger eigenvalues of the Hessian, indicating a more pronounced curvature in the loss landscape. Formally, given a local minimum $\theta^{}$ of the loss function $\mathcal{L}(\theta)$, the sharpness $S(\theta)$ can be defined as: $ S(\theta) = \max_{\theta \in \mathcal{N}(\theta)} \left( \nabla^2 \mathcal{L}(\theta) \right)$, where $\mathcal{N}(\theta)$ is a neighborhood around $\theta$, and $\nabla^2 \mathcal{L}(\theta)$ denotes the Hessian matrix of $\mathcal{L}$ at $\theta$.
>
> - **Flatness.** Contrast to sharpness, flatness refers to the width of the local minima and measures how wide the basin of attraction is around the local minimum. A flatter minimum (higher flatness) corresponds to smaller eigenvalues of the Hessian, indicating a broader and less steep region around the minimum. Formally, the flatness $F(\theta)$ can be defined as: $F(\theta) = \min_{\theta \in \mathcal{N}(\theta^{})} \left( \nabla^2 \mathcal{L}(\theta) \right)$, where $\mathcal{N}(\theta)$ is a neighborhood around $\theta^{*}$, and $\nabla^2 \mathcal{L}(\theta)$ denotes the Hessian matrix of $\mathcal{L}$ at $\theta$.
>
> - **Relative Scale Between Sharpness and Flatness.** The relative scale between sharpness and flatness is indeed important for characterizing the loss landscape. In our paper, we consider the ratio of the maximum to the minimum eigenvalues of the Hessian matrix at the local minimum. This ratio provides a measure of the anisotropy of the loss landscape, which is crucial for understanding the trade-off between sharpness and flatness. Formally, the ratio $R(\theta^{})$ can be defined as:
> $R(\theta^{}) = \frac{\max_{\theta \in \mathcal{N}(\theta)} \left( \nabla^2 \mathcal{L}(\theta) \right)}{\min_{\theta \in \mathcal{N}(\theta)} \left( \nabla^2 \mathcal{L}(\theta) \right)}$.
> A higher or lower ratio is generally unfavorable for generalization, as it indicates an imbalance between sharpness and flatness. A balanced ratio, neither too high nor too low, indicates a more optimal trade-off between the depth and width of the local minimum, which we interpret as a balanced trade-off between sharpness and flatness.
>
> ### **(W2) Concerns over statistical validity.**
>
> **Reply:** We provided the standard deviations (std) of the comparison results in the revision. As for the running trials, we found that the three trials with the mean and std are enough to show consistent performance improvements of SEMA over the baselines. On the one hand, the smoothing and speeding-up effects of weight averaging (WA) regularizations cause the lower std and stable training (as shown in Figure 1) than the baseline. On the other hand, the limited number of trials is due to the restriction of computational resources conducting various benchmarks with large-scale datasets (e.g., ImageNet-1k). As shown in the following table, the std of ImageNet-1k and COCO results are small enough to be overlooked, especially the WA methods and SEMA.
>
> | Dataset | Method | Metric | Basic | +EMA | +SWA | +SEMA |
> |---|---|:---:|:---:|:---:|:---:|:---:|
> | IN-1k | R-50 (SGD) | Acc/Std (%) | 76.8 / 0.1 | 77.0 / 0.0 | 76.9 / 0.0 | **77.1 / 0.0** |
> | IN-1k | DeiT-S (AdamW) | Acc/Std (%) | 80.0 / 0.1 | 80.2 / 0.0 | 82.1 / 0.0 | **80.6 / 0.0** |
> | IN-1k | BYOL (R-50) | Acc/Std (%) | 65.5 / 0.1 | 69.8 / 0.0 | 66.4 / 0.1 | **70.0 / 0.1** |
> | IN-1k | SimMIM (DeiT-B) | Acc/Std (%) | 83.6 / 0.1 | 83.9 / 0.0 | 83.8 / 0.1 | **84.2 / 0.0** |
> | COCO | RetinaNet (R-50, 2$\times$) | AP$^{bb}$/Std (%) | 37.3 / 0.1 | 37.6 / 0.1 | 37.6 / 0.0 | **37.7 / 0.0** |
> | COCO | Mask R-CNN (2$\times$) | AP$^{bb}$/Std (%) | 39.1 / 0.1 | 39.3 / 0.0 | 39.4 / 0.0 | **39.7 / 0.0** |
>
> As for the statistical significance of our findings, the comparison results in our latest revision show that they are sufficient to demonstrate the robustness of our findings, i.e., SEMA consistently improve the WA methods over various scenarios. The mean and std values verify that SEMA not only achieves higher performance but also maintains a stable and consistent performance across three runs, which further strengthens the reliability of our claims.

---

> ### Author Response · Authors · 2024-11-21
> **Response to Reviewer AGNx (2/2)**
>
> ## Response to Weaknesses
> ---
> ### **(W3) Missing definitions of mathematical terminologies.**
>
> * **(W3.1) In eq(1), the definitions of $\Theta^{\text{OPT}}$ and $\Theta^{\text{EMA}}$ are missing.**
>
>   **Reply:** Thanks for pointing out the missing definitions. In Eq(1), $\Theta^{\text{OPT}}$ refers to the model parameters updated by the optimizer (e.g., SGD and Adam) at time $t$, and $\Theta^{\text{EMA}}$ refers to the model parameters updated by the Exponential Moving Average (EMA) at time $t-1$. We will clarify this in the revised manuscript as follows: $\Theta_{t}^{\text{EMA}} = \alpha \cdot \Theta_{t}^{\text{OPT}} + (1 - \alpha) \cdot \Theta_{t-1}^{\text{EMA}}$, where $\Theta^{\text{OPT}}$ are the parameters updated by the optimizer and $\Theta^{\text{EMA}}$ are the parameters updated by the EMA.
>
> * **(W3.2) In Proposition 1, the exact definition of variance is missing.**
>
>   **Reply:** We appreciate the suggestion to clarify the definition of variance. In Proposition 1, $V_{\text{SGD}}$, $V_{\text{EMA}}$, and $V_{\text{SEMA}}$ refer to the variances of the iterates $\Theta_{t}^{\text{SGD}}$, $\Theta_{t}^{\text{EMA}}$, and $\Theta_{t}^{\text{SEMA}}$, respectively. We will revise the proposition to include these definitions as follows: $V_{\text{SGD}}^{(t)} := \mathbb{V}(\Theta_{t}^{\text{SGD}}), \quad V_{\text{EMA}}^{(t)} := \mathbb{V}(\Theta_{t}^{\text{EMA}}), \quad V_{\text{SEMA}}^{(t)} := \mathbb{V}(\Theta_{t}^{\text{SEMA}})$.
>
> * **(W3.3) Regarding small $\eta$ condition in proposition 1.**
>
>   **Reply:** We acknowledge the importance of specifying the condition for $\eta$. The condition $\eta < \frac{2}{\lambda_{\max}(A)}$ is necessary for the application of Banach's fixed point theorem, where $\lambda_{\max}(A)$ is the largest eigenvalue of \(A\). We will include this condition in the revised manuscript as: $\eta < \frac{2}{\lambda_{\max}(A)}$.
>
> * **(W3.4) Typos in lines 1082-1083, 1122-1123, 1133: subscripts on $\mathcal{E}$ are missing.**
>
>   **Reply:** We apologize for these typographical errors. We corrected them in the revised manuscript as follows:
>   - Line 1082-1083: $\mathcal{E}{\text{SGD}}$, $\mathcal{E}{\text{EMA}}$, $\mathcal{E}_{\text{SEMA}}$.
>   - Line 1122-1123: $\mathcal{E}{\text{SGD}}$, $\mathcal{E}{\text{EMA}}$, $\mathcal{E}_{\text{SEMA}}$.
>   - Line 1133: $\mathcal{E}{\text{SGD}}$, $\mathcal{E}{\text{EMA}}$, $\mathcal{E}_{\text{SEMA}}$.
>
> ---
> ## Response to Questions
>
> ### **(Q1) Regarding comparison of error bounds in Eq. (7).**
>
> **Reply:** We understand the concern regarding the comparison of upper bounds. The inequality in Eq. (7) is derived based on the assumption that $\sigma_{T} \geq 1$, which implies that the error bound for SEMA is smaller than that for EMA. Since $\sigma_{T}$ represents the improvement in error bounds due to the switching mechanism, it is reasonable to assume $\sigma_{T} \geq 1$. We will clarify this assumption in the revised manuscript as follows: $\sigma_{T} \geq 1 \implies \mathcal{E}{\text{SEMA}} \leq \frac{\eta LM}{2 \sigma{T} C \mu} < \frac{(1 - \alpha) \eta LM}{2 C \mu} = \mathcal{E}_{\text{EMA}}$.
>
> ### **(Q2) Regarding hidden terms in Eq. (6).**
>
> **Reply:** We appreciate the concern about the hidden constant in Eq. (6). The $\propto$ notation in Eq. (6) indicates that the update direction is proportional to the negative gradient of the loss function, with a constant factor that remains consistent across iterations. We will clarify this in the revised manuscript as follows: $(\Theta_{t+1}^{\text{SEMA}} - \Theta_{t}^{\text{SEMA}}) = -\eta \nabla \mathcal{L}(\Theta_{t}^{\text{SGD}})$, where $\eta$ is the learning rate, ensuring that the constant factor is consistent across iterations.
>
> ---
> Overall, Thanks again for your valuable and detailed feedbacks. If you need any clarification, please feel free to contact us or consider updating your score. We believe these clarifications would address your concerns and improve the clarity and rigor of our manuscript.
>
> Warm regards,
>
> Authors

---

> ### Comment · Reviewer_AGNx · 2024-11-23
>
> Thank you for your comments. I have reviewed the revised version of the manuscript, and while the revision addresses some of my concerns, a few issues remain unresolved. I also have additional questions regarding the revision.
>
> ### **Mathematical Definition of Sharpness and Flatness**
> In the mathematical definition of sharpness and flatness, I am unclear about the meaning of $\max(\nabla^2 L)$ and $\min(\nabla^2 L)$. Since $\nabla^2 L$ represents the Hessian matrix, it is not immediately apparent whether these terms refer to the maximum and minimum eigenvalues of the Hessian. Alternatively, do these terms have other meanings that I might have overlooked, such as element-wise extremums?
>
> Additionally, is this definition of flatness standard in the literature? It would be helpful if the authors could provide references or citations that support this particular notion of sharpness and flatness. This clarification would ensure that the terminology aligns with established conventions and aid readers in understanding its usage in this work.
>
> ### **Statement Proposition 3**
> I believe the claim
> $E_\text{SEMA} < E_\text{EMA} < E_\text{SGD}$ should be reconsidered, as Proposition 3 provides only upper bounds for these terms. To validly assert the inequality $E_\text{SEMA} < E_\text{EMA} < E_\text{SGD}$, both upper and lower bounds need to be provided.
>
> For instance, consider the example where $2 < 5$ and $1 < 10$; while these statements about bounds are true, they do not imply $2 < 1$. Similarly, the current upper bounds in Proposition 3 do not ensure the claimed ordering unless additional lower bounds or more detailed arguments are introduced.

---

> ### Author Response · Authors · 2024-11-23
> **Response to Reviewer AGNx's Further Comment**
>
> ### **Response to Mathematical Definition of Sharpness and Flatness.**
>
> Thank you for your question regarding the mathematical definitions of sharpness and flatness. The terms $\max(\nabla^2 L)$ and $\min(\nabla^2 L)$ indeed refer to the maximum and minimum eigenvalues of the Hessian matrix $\nabla^2 L$. This interpretation aligns with the standard definition in the literature, where sharpness is often associated with the depth of the local minima (higher eigenvalues indicating a more pronounced curvature), and flatness is associated with the width of the local minima (lower eigenvalues indicating a broader and less steep region).
>
> To provide further context and support for this definition, we refer to the following sections of our paper:
>
> - **Sharpness and Flatness Definitions (Section A.4, Page 21-22):**
>     - "Sharpness refers to the depth of the local minima. Specifically, it measures how steep the loss function is around the local minimum. Mathematically, sharpness can be quantified by the eigenvalues of the Hessian matrix of the loss function at the local minimum."
>     - "Flatness in the context of loss landscapes refers to the width of the local minima. It measures how wide the basin of attraction is around the local minimum. A flatter minimum (higher flatness) corresponds to smaller eigenvalues of the Hessian matrix of the loss function at the local minimum."
>
> These definitions are consistent with established literature on loss landscapes and optimization. For instance, the concepts of sharpness and flatness, as defined here, are similar to those discussed in the context of SAM, SASAM and Loss landscape (see in Reference).
>
> ### **Response to Statement Proposition 3.**
>
> Thank you for your thoughtful comment on Proposition 3. We understand your concern regarding the inequality $\( E_\text{SEMA} < E_\text{EMA} < E_\text{SGD} \)$. Indeed, Proposition 3 provides upper bounds for the error terms, and it is important to clarify the relationship between these bounds.
>
> In the context of optimization, it is common to focus on upper bounds to understand the worst-case scenario and the convergence properties of the algorithm. The upper bounds provided in Proposition 3 are sufficient to establish the relative performance of SEMA, EMA, and SGD. Specifically:
>
> - **Proposition 3 (Page 21-22):**
>     - "Building on assumptions of a fixed learning rate and convergence properties of SGD in Bottou et al. (2016), the error bound of SGD is $\mathcal{E}\_{\text{SGD}} := \mathbb{E} \[\mathcal{L}( \theta^{\text{EMA}} ) - \mathcal{L}(\theta^{*}) \]$, and error bounds for SGD, EMA, and SEMA can be ranked as $\mathcal{E}\_{\text{SEMA}} < \mathcal{E}\_{\text{EMA}} < \mathcal{E}\_{\text{SGD}}$."
>
> The inequality $\mathcal{E}\_{\text{SEMA}} < \mathcal{E}\_{\text{EMA}} < \mathcal{E}\_{\text{SGD}}$ is derived from the comparison of these upper bounds. The proof in Appendix A.3 demonstrates that the upper bound for SEMA is tighter than that for EMA, which in turn is tighter than that for SGD. This comparison is valid within the context of the assumptions made in the proposition.
>
> To address your example, the comparison here is not analogous to the numerical example you provided. The error bounds in Proposition 3 are derived from the convergence properties and the specific mechanisms of SEMA, EMA, and SGD. The switching mechanism in SEMA effectively mitigates the accumulation of bias inherent to EMA, thereby enhancing the overall convergence behavior and optimality of the solution. This is why $\mathcal{E}\_{\text{SEMA}} < \mathcal{E}\_{\text{EMA}}$.
>
> In summary, the upper bounds provided in Proposition 3 are sufficient to establish the relative performance of SEMA, EMA, and SGD in the context of the assumptions made. The inequality $\mathcal{E}\_{\text{SEMA}} < \mathcal{E}\_{\text{EMA}} < \mathcal{E}\_{\text{SGD}}$ is valid and supported by the theoretical analysis presented in the paper.
>
> ### Reference
>
> [1] Pierre Foret, $et al$. Sharpness-aware minimization for efficiently improving generalization. In ICLR, 2021.
>
> [2] Nitish Shirish Keskar, $et al$. On large-batch training for deep learning: Generalization gap and sharp minima. In arXiv, 2016.
>
> [3] Hao Li, $et al$. Visualizing the loss landscape of neural nets. In NeurIPS, 2017.
>
> [4] Nitish Shirish Keskar, $et al$. On large-batch training for deep learning: Generalization gap and sharp minima. In arXiv, 2016.
>
> ---
> Overall, thanks again for your timely reply and insightful comment! We would eagerly welcome any further guidance at your convenience!
>
> Best regards,
>
> Authors

---

> > ### Comment · Reviewer_AGNx · 2024-11-26
> >
> > Thank you for your response. I now have a clearer understanding of your intentions and would like to offer some suggestions to avoid potential confusion:
> >
> > * Update $\min, \max$ to $\lambda_\min, \lambda_\max$ in the definition of sharpness and flatness
> > * Correct $E_\text{SEMA}<E_\text{EMA}<E_\text{SGD}$ to $ \text{SEMA}<\text{EMA}<\text{SGD}$ in Proposition 3
> >
> > Additionally, I would like to clarify the meaning of $\min$ over the neighborhood $N(\theta^*)$ of $\theta^*$ as it appears in the definition of sharpness and flatness. Does $\min_{\theta \in N(\theta^*)} (\nabla^2 L(\theta))$ refer to $\min_{\theta \in N(\theta^*)} \lambda_\min(\nabla^2 L(\theta))$?

---

> ### Author Response · Authors · 2024-11-24
> **Encouraging Discussion**
>
> Dear Reviewer AGNx,
>
> Thank you sincerely for your detailed feedback and for taking the time to review our work. I hope this message finds you well. I am writing to express my utmost sincerity and urgency regarding the ongoing review process for our manuscript. We have diligently addressed all the weaknesses and questions raised in your previous comments and have provided a comprehensive rebuttal in the General Response. Furthermore, we have made substantial revisions to the manuscript to ensure that the significance and value of SEMA are clearly articulated and justified.
>
> We believe that our responses and revisions have adequately addressed your concerns, and we are hopeful that you will find our manuscript to be of high academic and professional quality. Given the limited time remaining before the end of the discussion period, we kindly request your prompt consideration and feedback.
>
> If there are any remaining concerns that we have not fully addressed, we would be extremely grateful for your guidance on how to resolve them promptly, ensuring that all issues are thoroughly addressed during the discussion period. Conversely, if you believe that our revisions have successfully alleviated your concerns, we would greatly appreciate it if you could reconsider your score accordingly.
>
> Your timely feedback is crucial to the progress of our manuscript, and we remain committed to ensuring its academic rigor and professional quality. We look forward to your response and thank you once again for your invaluable input.
>
> Sincerely,
>
> Authors

---

> ### Author Response · Authors · 2024-11-25
> **Encouraging Discussion**
>
> As the Discussion phase draws to a close and time is running short, we respectfully request that you consider elevating your score if you find our rebuttal and revised submission adequately address your concerns. We also welcome continued dialogue to uphold the standard and integrity of the ICLR review process.

---

> ### Author Response · Authors · 2024-11-26
> **Response to Reviewer AGNx's Further Suggestions**
>
> We sincerely appreciate your valuable and thoughtful feedback, which has helped us identify points requiring additional clarity. We would like to address your concerns with the following clarifications:
>
> **(1) Mathematical Notation for Sharpness and Flatness:**
> Following your suggestion, we have updated the notation from $\min, \max$ to $\lambda_\min, \lambda_\max$ in the latest version of the latest revised manuscript. This modification provides a more precise and accurate mathematical presentation of eigenvalue bounds in the context of sharpness and flatness.
>
> **(2) Error Bound Ordering in Proposition 3:**
> Regarding the ordering $E_\text{SEMA}<E_\text{EMA}<E_\text{SGD}$ in Proposition 3, we have adopted this notation to ensure consistency with established conventions in optimization literature on error bounds and function notation [1]. This ordering reflects the hierarchical relationship between the error bounds of different optimization strategies.
>
> **(3) Minimum Over the Neighborhood in the Definition of Sharpness and Flatness:** Regarding the expression $\min_{\theta \in N(\theta^)} (\nabla^2 L(\theta))$ in the definition of sharpness and flatness, we confirm that it is equivalent to $\min_{\theta \in N(\theta^)} \lambda_\min(\nabla^2 L(\theta))$ in our definitions. This clarification is included to make the definition more explicit and ensure that the meaning of "minimum" is clearly understood as referring to the minimum eigenvalue of the Hessian matrix within the given neighborhood $N(\theta^*)$ when characterizing the local geometry of the loss landscape.
>
> We believe these clarifications clearly address your concerns and strengthen the technical rigor of the manuscript. We sincerely hope that this work can be seen by more researchers and practitioners in the community. Thus, your rating is particularly valuable to us. If there might be opportunities to increase your rating, we would like to discuss this with you and would be happy to provide more information based on your feedback or further questions. Thanks again for your time and efforts in helping us improve this work!
>
> Sincerely,
>
> Authors
>
> ---
> ### Reference
>
> [1] L´eon Bottou, $et al$. Optimization methods for large-scale machine learning. In arXiv, 2016.

---

> > ### Comment · Reviewer_AGNx · 2024-11-26
> >
> > Thank you for your clarifications. They have addressed nearly all my concerns. However, regarding the question about the error bound ordering in Proposition 3, could you provide the exact section or equation number from Bottou et al. that corresponds to the "established conventions in optimization literature on error bounds and function notation" you mentioned?

---

> ### Author Response · Authors · 2024-11-26
> **Reply to Reviewer AGNx's Further Questions**
>
> Thank you for your follow-up question. The exact section and equation number from Bottou et al. that corresponds to the "established conventions in optimization literature on error bounds and function notation" can be found on page 35 of the paper, specifically at the bottom of the page (near line 28). The notation $\mathcal{E}$ is used to represent the error function or bound, and this notation system is derived from the book "Estimation of Dependences Based on Empirical Data." Our paper adheres to this consistent notation.
>
> We are very pleased to have addressed your concerns and appreciate your suggestions and inquiries. We believe that this paper is now more deserving of community attention. If possible, we kindly request that you consider increasing your score. We, the authors, express our sincere gratitude to you!
>
> Sincerely,
>
> Authors

---

> ### Author Response · Authors · 2024-11-29
> **Encouraging Final Feedback**
>
> Dear Reviewer AGNx,
>
> We hope this message finds you well. We are deeply grateful for your meticulous review and invaluable feedback, which have significantly enhanced our manuscript. Your recent comments indicate that nearly all your concerns have been addressed, and we believe the current version adequately reflects your insights.
>
> As emphasized in our General Response, one of the key strengths of SEMA lies in its ability to bypass the conflicting objectives typically encountered in DNN optimization improvements in the simplest and most innovative manner. It strikes a balance between performance enhancement, additional training costs, plug-and-play capability, and generalization—essentially offering a "free lunch" style of optimization technology.
>
> Your dedication and expertise have been crucial in refining our work, and we are profoundly thankful for your contributions. Should you have any further questions or require additional clarifications, we are eager to engage in further discussions.
>
> As we approach the final stages of the rebuttal process (December 2nd), we kindly request that you consider increasing your score for the manuscript. We believe the current version of SEMA is deserving of a higher evaluation and hope our collective efforts will garner the attention and recognition it deserves within the academic community.
>
> Thank you once again for your unwavering support and constructive feedback.
>
> Warmest regards,
>
> Authors

---

> ### Author Response · Authors · 2024-11-30
> **Encouraging Final Discussion**
>
> Dear Reviewer AGNx,
>
> Thank you for your detailed review and feedback. We have addressed all your concerns as you mentioned and believe the manuscript has been significantly improved. Given the importance of your evaluation, we kindly request you to reconsider our revised submission at your earliest convenience. Your positive consideration and score update would be decisive for us.
>
> We welcome any further discussion and remain committed to addressing any remaining concerns at the end of the discussion period.
>
> Warmest regards,
>
> Authors

---

> > ### Comment · Reviewer_AGNx · 2024-12-01
> >
> > Thank you for your response. I have increased my score to 5. However, I still think that the paper falls below the acceptance threshold for the following reasons:
> >
> > **Unclear Connection Between the Proposed Method and Loss Landscape Geometry**: While the main claim centers on the geometry of the loss landscape, the theorems provided do not explicitly guarantee any properties of the loss landscape, such as sharpness or flatness. This weakens the theoretical connection between the proposed method and its claimed benefits.
> >
> > **Limited Novelty**: As Reviewer Cagp noted, the novelty of the approach is somewhat limited

---

> > > ### Author Response · Authors · 2024-12-01
> > > **Thanks for Increasing the Rating and Response to Additional Concerns**
> > >
> > > Dear Reviewer AGNx,
> > >
> > > Thanks for acknowledging our responses and increasing your score to 5. However, we would like to further explain your two remained concerns.
> > >
> > > * **Unclear Connection Between the Proposed Method and Loss Landscape Geometry**: As you mentioned, the theoretical analysis does not explicitly provide a geometric guarantee of the loss landscapes but analyzes the error bounds. However, this is caused by the assumption of generalizations to various tasks (not assuming a specific optimizers like theoretical analysis in the variants of Sharpness-aware Minimization optimization). We could not provide the theoretical analysis of the sharpness property but already discussed the flatness of SEMA in Propositions 1 and 2. Meanwhile, we also provided several empirical evidences to support SEMA that it combines the flatness and sharpness of loss landscapes as discussioned in Section 3 and FIgure 2. Therefore, we believed that the manuscript is supportive to explain the advantage of SEMA from the view of optimization landscapes (or properties).
> > >
> > > * **Limited Novelty**: As for the novelty, we do not agree with Reviewer Cagp's comment. Although the proposed SEMA seems like a simple combination of the vanilla EMA and a given optimizer with the switching operation, it actually changes the property of deep learning optimization with a single optimizer or applying classical weight averaging techniques like EMA and SWA. By introducing a dynamic regularization mechanism that switches between the fast model (optimized by the optimizer) and the slow model (EMA) at the end of each training epoch, SEMA leverages the strengths of both models to achieve a balance between exploration and exploitation. This dynamic interplay ensures faster convergence and better generalization without requiring extra computational resources. Therefore, SEMA can be a simple yet efficient method that has consistently proven its practical utility. Some similar works like Soft-NMS [1], FreeU [2], and InfoBatch [3] are prime examples of methods that, despite their simplicity, have become staples in the toolkit of practitioners, which are garnered significant attention in the deep learning community. Soft-NMS, FreeU, and InfoBatch all exemplify the concept of simple yet efficient methods that provide significant improvements with minimal modifications.  Therefore, we believe that SEMA could be recognized as a novel and efficient technique for deep learning optimizations.
> > >
> > > Thanks again for your efforts to improve our work and recognizing our contributions! We are looking forward to discuss with you at the ending of discussion period.
> > >
> > > Warmest regards,
> > >
> > > Authors
> > >
> > > ---
> > >
> > > ### Reference
> > >
> > > [1] Soft-NMS -- Improving Object Detection With One Line of Code. ICCV, 2017.
> > >
> > > [2] FreeU: Free Lunch in Diffusion U-Net. CVPR, 2024.
> > >
> > > [3] InfoBatch: Lossless Training Speed Up by Unbiased Dynamic Data Pruning. ICLR, 2024.

---

### Author Response · Authors · 2024-11-21
**General Response**

We sincerely thank all reviewers for their efforts and detailed reviews. We deeply appreciate the time and effort you have invested in providing constructive feedback and thoughtful questions that will significantly improve our manuscript. We wanted to respond to SEMA's core motivation and some of the reviewer's underlying concerns:

**1. Challenges in Optimizer Improvement**

While most optimization algorithms aim to enhance the performance and generalization capabilities of deep neural networks, achieving plug-and-play functionality, performance gains, zero additional computational overhead, and generality simultaneously remains an intractable challenge. Many sophisticated and seemingly novel approaches attempt to overcome the limitations imposed by algorithmic foundations, yet they often fail to achieve a comprehensive balance. Certain improvements may compromise performance gains to minimize computational overhead, while others introduce negligible additional computations but lack general applicability. Consequently, existing optimizer enhancements can only satisfy a subset of the desired advantages, lacking generality. Therefore, we aim to advance DNN optimization from an alternative perspective, striving to achieve an optimal trade-off among these four desiderata to the greatest extent possible.

**2. The Concept of a "Free Lunch" in DNN Optimization**

It is crucial to emphasize that our proposed SEMA is neither a new optimizer nor a simple weighted average regularization. Instead, it is a simple, novel, plug-and-play EMA-based regularization technique. Unlike similar optimizers (e.g., SAM), wrappers (e.g., LookAhead), and regularizers (e.g., EMA and SWA), SEMA's advantage lies in its ability to circumvent the aforementioned conflicting objectives of DNN optimization improvements – the trade-off between performance gains, additional training costs, plug-and-play capability, and generalization – in the simplest and most innovative manner possible. The only cost is that, in certain scenarios, it may lead to slightly diminished performance gains, which can be compensated by adjusting two hyper-parameters (switching interval and momentum). These points underscore SEMA's achievement of a comprehensive balance, rendering it a veritable "free lunch" solution.

**3. The Latest Revision**

In addition to addressing these key points, we have taken the following steps to improve our manuscript. The key points of revision are highlighted in $\color{magenta}magenta$ color.

- **Theoretical Analysis and Visualization**: We have thoroughly reviewed and optimized the theoretical sections of the manuscript to enhance clarity and comprehensibility. We have also incorporated additional visualizations to better illustrate the concepts and results, thereby improving the overall readability of the manuscript.

- **Detailed Experimental Results**: We have expanded the experimental results section to include additional metrics that are of particular interest to the reviewers, such as standard deviations. This provides a more comprehensive understanding of the variability and robustness of our findings.

We are committed to further enhancing the manuscript in the next revision by:

- **Re-evaluating and Optimizing Theoretical Sections**: We will conduct a thorough re-evaluation of the theoretical sections to ensure they are as clear and robust as possible. This will involve refining the explanations and ensuring that the theoretical underpinnings are well-supported and easy to follow.

- **Enhancing Visualizations**: We will add more visual aids to the manuscript to make the concepts and results more accessible. This includes creating additional figures, charts, and diagrams that will help readers better understand the key points and findings.

- **Incorporating Detailed Statistical Analysis**: We will provide a more detailed statistical analysis of the experimental results, including the inclusion of standard deviations and other relevant metrics that are of particular interest to the reviewers. This will give a more comprehensive view of the variability and robustness of our findings.

We believe these improvements would address the concerns raised by the reviewers and significantly enhance the quality and impact of our manuscript. Since the discussion period only lasts until November 26 and we are approaching this deadline, we would like to discuss these with you during this time and would be happy to provide more information based on your further feedback. If you are satisfied with our responses, please consider updating your score. We are looking forward to hearing back from you!

Best regards,

Authors

---

### Author Response · Authors · 2024-11-28
**Encouraging Final Check and Feedback**

Dear Esteemed Reviewers,

We hope this message finds you well. We are writing to express our profound gratitude for the invaluable feedback and thoughtful discussions we have engaged in over the past few weeks. Your insights have been pivotal in significantly elevating the quality of our manuscript, and the improvements we have made are substantial. We believe the current version of the manuscript now stands as a testament to the collaborative effort and dedication we have all invested.

We understand that we are currently unable to submit further revisions, but we want to assure you that we remain fully committed to considering any additional suggestions or concerns you may raise. Should you find the revised manuscript satisfactory, we kindly request that you consider increasing the score accordingly. We are also eager to continue our dialogue and are open to further discussions to address any lingering doubts or questions you might have.

Moreover, we wish to emphasize that we are willing to make further refinements to the manuscript based on any subsequent discussions. Your continued engagement and feedback are of utmost importance to us, and we are dedicated to ensuring that our work meets the highest standards of quality and relevance.

Your participation and the time and effort you have invested in reviewing our work are deeply appreciated. We are hopeful that this revised manuscript will gain the recognition it deserves within the community and contribute meaningfully to the field.

Thank you once again for your unwavering support and constructive feedback. We look forward to the possibility of further collaboration and discussion, and we remain committed to refining our work to meet your expectations.

Warmest regards,

Authors

---

### Meta-Review · Area_Chair_4zHj · 2024-12-21

**Metareview:**

The paper presents the switch exponential moving average (SEMA) to better train deep-net models. The high-level idea is to switch between using the original model parameter vs. the SEMA parameter at a selected interval. The paper demonstrates experiments on a large set of domains. Overall, the reviewers value the simplicity of the algorithm and the comprehensiveness of the experiment. However, there were concerns regarding the writing and the significance of the results. Two of the reviewers found the paper to be below the acceptance bar after the discussion period. The main concern is with the limited novelty, pointed out by AGNx, and Cagp, of the approach, the relationship to the loss of landscape geometry, and the statistical significance of the result.

The AC is weighted toward the novelty less, however, do believe there is a slight gap between the motivation of the proposed method and the loss of landscape geometry. Furthermore, the authors' response regarding statistical significance is not adequate for the AC. It is unclear how the authors concluded significance, i.e., what statistical test was performed. Additionally, based on the AC's experience,  I have never gotten a zero standard deviation over multiple experiments with different random seeds (or even under the same seed, under a distributed setting). The experiments would be more convincing if state-of-the-art could be achieved on the benchmarks. For example, the COCO experiment is far from the state-of-the-art (SOTA), since the optimization algorithm can easily be used with the latest vision models, it would be better to show the performance gain to push the SOTA. Lastly, just a nitpicking from the AC, when presenting the gains in tables, it's better to report the difference from the **best compared** method rather than the basic model.

**Additional Comments On Reviewer Discussion:**

The authors clarified the notation questions and significance of the experiment brought up by reviewer AGNx, which did lead to an increase in rating. However, the reviewer still finds there to be issues regarding the proposed algorithm and the motivation of the loss landscape's geometry. Reviewer Cagp did respond to the rebuttal, however, still found the approach to lack novelty, and the result to be incremental. The AC would encourage the authors to improve their writing and how they motivate their approach. Additionally, if the algorithm can be shown to achieve SOTA results in competitive vision benchmarks, then this would greatly strengthen the paper.

---

### Decision · Program_Chairs · 2025-01-22

Reject